# OSPA: Enhancing Identity-Preserving Image Generation via Online Self-Preference Alignment

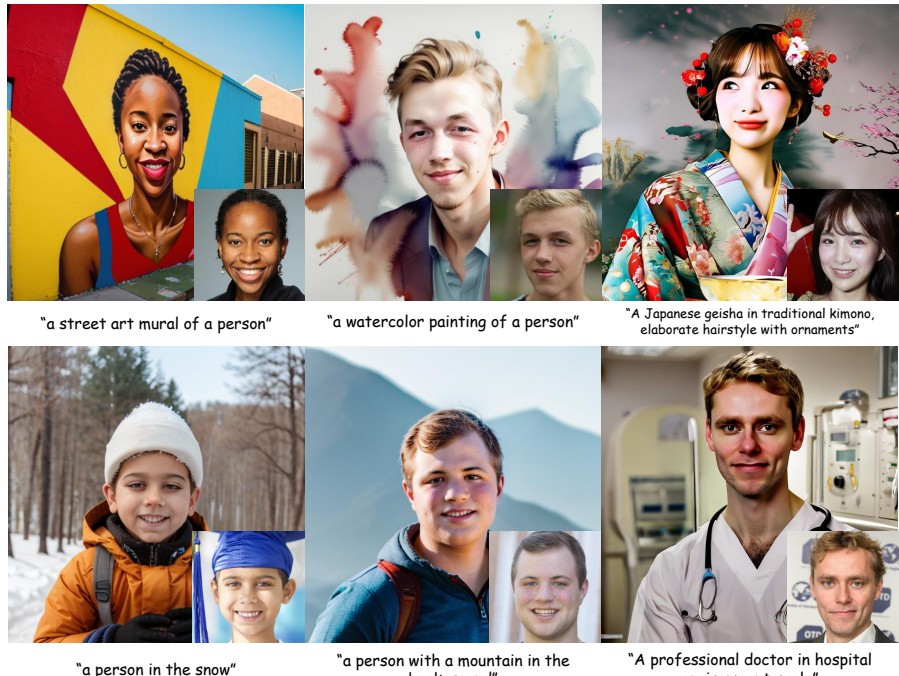

"a street art mural of a person"   "a watercolor painting of a person"   "A Japanese geisha in traditional kimono, elaborate hairstyle with ornaments"

"a person in the snow"   "a person with a mountain in the background"   "A professional doctor in hospital environment, male"

Figure 1: OSPA enhances identity fidelity and visual quality in images by aligning self-preference in an online manner.

## Abstract

Identity-preserving text-to-image generation has recently received increasing attention, yet it remains a challenging task. Existing approaches typically fine-tune diffusion models, but they often fail to preserve identity information reliably. Reinforcement learning with human feedback (RLHF) can improve identity consistency, but it requires expensive reward models and curated annotations, limiting its applications. We present Online Self-Preference Alignment (OSPA), a plug-and-play framework that achieves identity-preserving generation without relying on external reward models or high-quality datasets. OSPA exploits self-preference signals through three components: (1) a self-preference sample generation module that perturbs a frozen policy model to produce paired samples with explicit preferences; (2) a self-reward preference optimization mechanism that performs self-reward assessment and updates the policy using group preference optimization; and (3) an online curriculum learning strategy that continuously refines the sample generator with feedback from the evolving policy model. Comprehensive experiments on four state-of-the-art identity-preserving models demonstrate that OSPA substantially improves identity fidelity while maintaining visual quality, offering a general and effective alignment strategy for generative models.

# 1 INTRODUCTION

Identity-preserving image generation aims to synthesize new images of a specific individual from free-form text descriptions while faithfully maintaining facial identity. This task has received increasing attention in both academia and industry due to its wide applicability in e-commerce (Xue et al., 2024), personalized content creation (Wu et al., 2024).

Previous methods primarily finetune the pretrained diffusion models (Rombach et al., 2022; Podell et al., 2023) using a handful of reference images, such as Instantbooth (Shi et al., 2024), ELITE (Wei et al., 2023), Subject-Diffusion (Ma et al., 2024). Other methods leverage a face encoder to extract facial identity features and fine-tune the adapter, such as PhotoMaker (Li et al., 2024b), IP-Adapter (Ye et al., 2023), and InstantID (Wang et al., 2024). Inspired by the success of preference learning in large language models (LLM), recent works have explored feedback-driven optimization for image generation (Black et al., 2023; Xu et al., 2023; Zhang et al., 2024), which train reward models or use curated preference datasets to guide diffusion models. In the identity-preserving setting, ID-Aligner (Chen et al., 2024) introduces specialized rewards for consistency and aesthetics. All these methods achieve commendable results.

However, despite these achievements, these methods still fall short in several aspects, as depicted in Fig. 2. (1) Identity-preserving image generation methods based on supervised fine-tuning typically employ a mean squared error (MSE) loss during training, which is unable to explicitly learn image generation that faithfully captures the characteristics of the reference portrait. In addition, these methods **lack feedback** from rewards or humans, often resulting in the generation of preference-agnostic images, as shown in Fig. 2 (a). (2) Identity-preserving image generation methods based on reward tuning always **require a tailored reward model trained with human-annotated preference feedback data**, and encounter inefficient optimization caused by the offline feedback with the pre-collected preference datasets (see Fig. 2 (b)). (3) Both of the aforementioned methods largely overlook the **inherent stochasticity of generative models**, leading to significant variations in the quality distribution of the generated images (see the blue line in Fig. 2 (c)). However, this very property **offers a theoretical foundation for self-reward alignment**: by simply guiding the model to optimize in the direction of its own preferences, we can potentially enhance the consistency of the generated identities (see the red line in Fig. 2 (c) and Fig. 2 (d)).

In response to these challenges and findings, we propose Online Self-Preference Alignment (OSPA), a plug-and-play framework that enhances identity fidelity without relying on reward models or curated preference datasets. As illustrated in Fig. 3, OSPA consists of three collaborative modules. Specifically, we first propose a self-preference sample generation module, which creates self-sufficient preference pairs via self-perturbation, enabling continuous preference generation without requiring any external models or human participation. Then, we present a self-reward preference optimization module that addresses suboptimal results caused by misaligned feedback from external rewards by providing stable policy updates through self-preference scoring. Finally, an online curriculum preference learning module, which continually refines the policy model for broader exploration. By integrating these components, OSPA delivers consistent improvements in identity preservation under preference learning.

The main contributions are summarized as follows: (i) We provide new empirical insights into identity-preserving generation, showing that the inherent stochasticity of generative models offers a theoretical foundation for self-reward alignment. (ii) We propose Online Self-Preference Alignment (OSPA), a plug-and-play framework that removes reliance on external reward models or curated preference datasets, and leverages self-generated preference signals for stable online optimization. (iii) We conduct extensive experiments with representative methods, demonstrating that OSPA consistently improves identity fidelity while preserving visual quality.

# 2 RELATED WORK

## 2.1 IDENTITY-PRESERVING IMAGE GENERATION

ID-preserving image generation (Chen et al., 2023; Li et al., 2024b; Shi et al., 2024; Wang et al., 2024; Yan et al., 2023; Ye et al., 2023) has become an important branch of text-to-image generation, aiming to synthesize novel images of a given identity guided by prompts. Unlike conventional text-

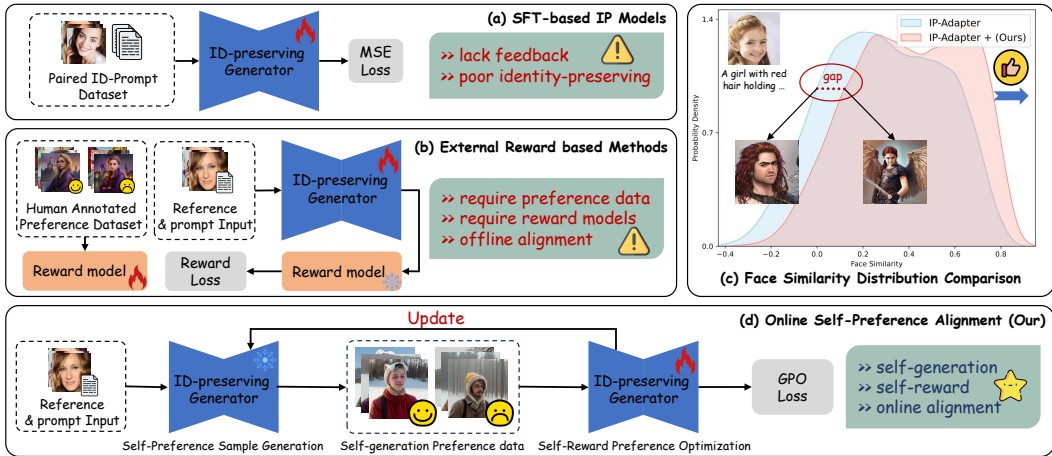

Figure 2: Comparison of the existing ID generation methods and the proposed OSPA. (a) ID models based on supervised fine-tuning lack feedback and suffer from poor identity preservation (see the blue line in (c)); (b) Reward-based methods rely on reward models and high-quality preference data, and are unable to align the model's preferences in real time; (c) The inherent stochasticity of generative models results in substantial variations in the quality distribution of generated images (the face similarity scores ranging from –0.4 to 0.8). (d) Our OSPA exploits this inherent property to generate self-preference data, perform self-reward evaluation, align the model's preferences in an online manner, and finally enhance identity consistency (see the red line in (c)).

to-image tasks, it requires both high-quality generation and strict consistency with the reference identity. Early few-shot methods (Wei et al., 2023; Shi et al., 2024) fine-tuned diffusion models on reference images, but such per-identity tuning restricts flexibility. To address this, subsequent works developed more efficient strategies: PhotoMaker (Li et al., 2024b) modifies image encoder layers to merge class and identity embeddings; IP-Adapter-FaceID (Ye et al., 2023) leverages face recognition embeddings; and InstantID (Wang et al., 2024) integrates semantic face embeddings and spatial control. Different from these methods, our approach adopts direct preference optimization, avoiding complex network designs while achieving versatile and effective ID preservation.

### 2.2 DIFFUSION METHODS WITH PREFERENCE OPTIMIZATION

Building on the success of reinforcement learning with human feedback (RLHF) in large language models (Ouyang et al., 2022), researchers have explored preference optimization in text-to-image generation. Representative works include DDPO (Su et al., 2024), which aligns diffusion models via reinforcement learning, ImageReward (Xu et al., 2023), which applies reward-based fine-tuning, backpropagates and updates the diffusion model by reward score, and Diffusion-DPO (Wallace et al., 2024), which extends DPO to diffusion models. More recently, several studies (Wallace et al., 2024; Ren et al., 2024; Xue et al., 2025) have adapted preference drawing inspiration from LLMs to improve image generation, while ID-Aligner (Chen et al., 2024) introduces a feedback-based framework for identity-preserving tasks. However, reward models are costly to train and prone to over-optimization issues such as mode collapse. Moreover, DPO-based methods remain underexplored for ID-preserving generation. To address these gaps, we propose a direct preference optimization variant that explicitly incorporates identity preservation into the alignment objective.

### 3 METHODOLOGY

#### 3.1 PROBLEM ANALYSIS AND FORMULATION

During the diffusion process of SFT-based identity-preserving text-to-image generation, noise $\epsilon$ is sampled and added to the data sample $x_0$ based on a predefined noise schedule. Then, a denoising model $\epsilon_\theta$ predicts the added noise based on the additional conditions $C$ and the noisy sample $x_t$. Once the model $\epsilon_\theta$ is trained, images can be generated from the random noise in an iterative manner.

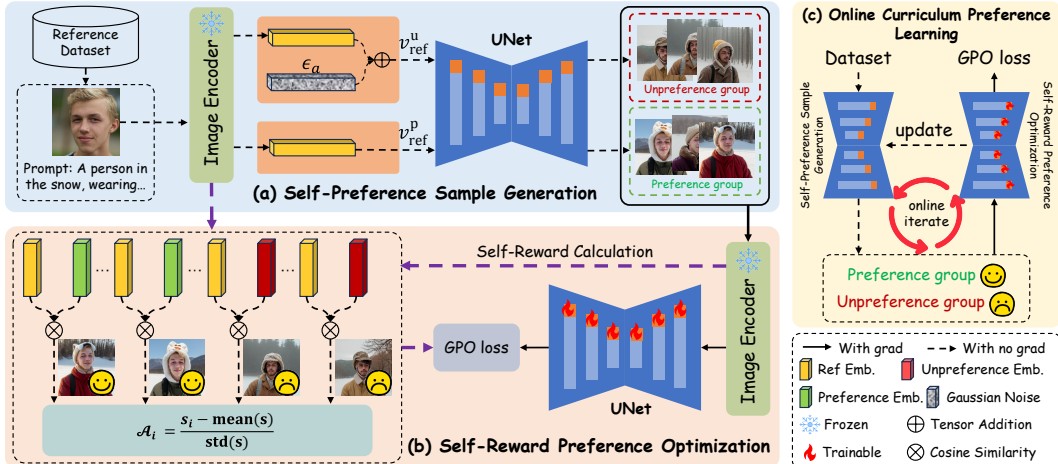

Figure 3: Overview of OSPA. (a) Self-preference samples are generated by perturbing image embedding with the Gaussian noise $\epsilon_a$ using the frozen ID policy model. (b) The reward score $\mathcal{A}_i$ for self-reward preference optimization is computed as the similarity between the self-preference samples and the reference image. (c) We update the ID policy model through online curriculum preference learning to continuously refine both the model and the generated preference data.

Therefore, it is natural to randomly generate images with obvious differences in ID preservation, where $x_i^p \in \{x_1^p, x_2^p, \dots, x_n^p\}$ is the preference generation image, and $x_i^u \in \{x_1^u, x_2^u, \dots, x_n^u\}$ is the unpreference generation image. This property naturally fits the reinforcement learning methods (Black et al.; Lee et al., 2023), which refine generation models through ranked preference pairs. However, the RLHF-based identity-preserving text-to-image generation method (Chen et al., 2024) relies on external rewards—specifically, identity consistency and identity aesthetic rewards—which can lead to suboptimal results and necessitate a large amount of human-annotated preference data.

This issue raises a key question: Instead of relying on external models, can we enable the model to learn its own preferences from its generation results—guiding it toward high-quality outputs that it inherently "approves" of? To this end, we formally introduce the self-reward generation paradigm, which replaces external reward models with the model's own confidence and uncertainty—enabling self-validation that even elicits latent endogenous rewards already present in the generative model. In this work, we propose the first instantiation of the self-reward generation paradigm—namely, the Online Self-Preference Alignment (OSPA) model—which performs real-time self-preference modeling and optimization during generation, enabling continuous, "self-driven" improvement without relying on any external auxiliary modules.

## 3.2 ONLINE SELF-PREFERENCE ALIGNMENT

**Overview.** Given a face-specific paired image-text dataset $\mathcal{D}_{ref} = (\mathcal{P}, \mathcal{X}_{ref})$, where $\mathcal{P}$ is a set of prompts and $\mathcal{X}_{ref}$ is a set of the reference images, our objective is to break the dependence on high-quality preference data and auxiliary reward models, and boost the alignment between identity preservation and the model's self-preferences. Fig. 3 presents the proposed Online Self-Preference Alignment (OSPA), which is a novel and plug-and-play identity-preserving text-to-image generation framework. It consists of three crucial components, namely Self-Preference Sample Generation, Self-Reward Preference Optimization, and Online Curriculum Preference Learning, to facilitate more effective preference optimization.

**Self-Preference Sample Generation.** The preference dataset employed in reward modeling or reinforcement learning is collected by manual annotation or automatic construction based on external models. The first one is labor-intensive and expensive, and lacks standardized qualitative criteria for assessing facial similarity. This prevents the model from receiving precise preference feedback and leads to unsustainable learning due to reliance on additional datasets. To tackle this issue, we introduce the Self-Preference Sample Generation module, as shown in Fig. 3 (a), to facilitate more effective preference generation without manual or additional model involvement.

---

**Algorithm 1:** Online Self-Preference Alignment.

---

**Input:** Reference Set $\mathcal{D}_{ref} = (\mathcal{P}, \mathcal{X}_{ref})$, Pre-trained Policy Model $\mathcal{G}(\cdot)$, Image Encoder $\mathcal{E}_I$
**Output:** Online Self-Preference Alignment $\mathcal{G}_\theta$

1 **for** $x_{ref}$ *in* $\mathcal{D}_{ref}$ **do**
2      // Self-Preference Sample Generation.
3      $\boldsymbol{v}_{\text{ref}}^p = \mathcal{E}_I(x_{\text{ref}}), \boldsymbol{v}_{\text{ref}}^u = \mathcal{E}_I(x_{\text{ref}}) + \alpha\epsilon_a, c \sim \mathcal{P}$
4      $\{x_{\text{gen}}^{p_1}, x_{\text{gen}}^{p_2}, \cdots, x_{\text{gen}}^{p_n}\} \sim \mathcal{G}_\theta(\boldsymbol{v}_{\text{ref}}^p, c)$
5      $\{x_{\text{gen}}^{u_1}, x_{\text{gen}}^{u_2}, \cdots, x_{\text{gen}}^{u_m}\} \sim \mathcal{G}_\theta(\boldsymbol{v}_{\text{ref}}^u, c)$
6      // Self-Reward Preference Calculation.
7      **for** $i$ *in* $x_{gen}^{p_i}$ **do**
8          $S_i = \text{sim}(\boldsymbol{v}_{\text{ref}}, \mathcal{E}_I(x_{\text{gen}}^{p_i}))$
9      **for** $j$ *in* $x_{gen}^{u_j}$ **do**
10         $S_{j+n} = \text{sim}(\boldsymbol{v}_{\text{ref}}, \mathcal{E}_I(x_{\text{gen}}^{u_j}))$
11      // Online Preference Optimization.
12      Compute $\mathcal{A}_i$ from $S$ for $\mathcal{L}_{\text{GPO}}$
13      $\mathcal{G}_\theta \leftarrow \mathcal{G}_\theta + \nabla_{\mathcal{G}_\theta} \mathcal{L}_{\text{GPO}}$

---

Specifically, given the reference face image $x_{ref}$, we first employ the image encoder $\mathcal{E}_I(\cdot)$ (such as the face encoder in IP-Adapter (Ye et al., 2023)) from the frozen ID policy model $\mathcal{G}$ to extract the image embeddings. We then randomly perturb a subset of image embeddings with Gaussian noise $\epsilon_a \in \mathcal{N}(0, I)$ to generate preferred and unpreferred face embeddings as follows:

$$\boldsymbol{v}_{\text{ref}}^p = \mathcal{E}_I(x_{\text{ref}}) \tag{1}$$

$$\boldsymbol{v}_{\text{ref}}^u = \mathcal{E}_I(x_{\text{ref}}) + \alpha\epsilon_a \tag{2}$$

where $\alpha$ is the noise intensity coefficient, $\boldsymbol{v}_{ref}^p$ is the preferred face embedding, and $\boldsymbol{v}_{ref}^u$ is the unpreferred face embedding. This enables us to generate the preferred sample $x_{gen}^p$ and the unpreferred sample $x_{gen}^u$ using ID policy model $\mathcal{G}$ as:

$$x_{\text{gen}}^p = \mathcal{G}_{\text{DDIM-sample}}(\boldsymbol{v}_{\text{ref}}^p, c) \tag{3}$$

$$x_{\text{gen}}^u = \mathcal{G}_{\text{DDIM-sample}}(\boldsymbol{v}_{\text{ref}}^u, c) \tag{4}$$

where $c$ is the prompt sample from $\mathcal{P}$. Finally, we generate multiple images $\mathcal{X} = \{x_{\text{gen}}^1, x_{\text{gen}}^2, \ldots, x_{\text{gen}}^G\}$ with explicit self-preference. In this way, the ID policy model receives accurate feedback aligned with its self-preferences, leading to a more effective alignment process.

**Self-Reward Preference Optimization.** The preference generation method (Chen et al., 2024) fine-tunes the ID generation model by leveraging reward model training with human-annotated preference data. Although using external reward models offers an intuitive approach, it suffers from the following shortcomings: Training reward models requires large-scale preference data, which is expensive and time-consuming to collect; Due to offline updates, the trained reward model may become outdated, leading to progressive misalignment between the reward model and the policy model; Since the reward model and policy model are distinct, an additional information gap arises during preference training, leading to suboptimal results.

To address these problems, an intuitive idea is to determine the preference through the policy model itself, so we proposed Self-Reward Preference Optimization mechanism, as shown in Fig. 3 (b). Specifically, we also use the image encoder $\mathcal{E}_I(\cdot)$ to extract the image embedding $\boldsymbol{v}_{\text{gen}}^i$ for the generated sample $x_{\text{gen}}^i$. Then, the identity similarity $S_i$ between $\boldsymbol{v}_{\text{gen}}^i$ and $x_{\text{gen}}^i$ is is defined as,

$$S_i = \text{sim}(\boldsymbol{v}_{\text{ref}}, \boldsymbol{v}_{\text{gen}}^i), i \in \{1, 2, \ldots, G\} \tag{5}$$

To ensure stable optimization of preference generation, we utilize the Group Preference Optimization (GPO) (Chen et al., 2025), which fine-tunes the policy model to maximize the rewards of the entire group, and the objective is defined as:

$$\mathcal{L}_{\text{GPO}} = \mathbb{E}_{t \sim H(1,T), x_0 \sim p(x_0), \epsilon \sim \mathcal{N}(0,1)} \sum_{i=1}^{G} [\mathcal{A}_i(\|\epsilon - \epsilon_\theta(x_t^i, t)\|_2^2 - \|\epsilon - \epsilon_{ref}(x_t^i, t)\|_2^2)] \tag{6}$$

Table 1: Quantitative comparison between the state-of-the-art methods and those with our method. The best results are highlighted in bold. Sim. refers to the face similarity score.

| Architecture | Methods | Sim.↑ | CLIP-I↑ | FLIP-I↑ |
|---|---|---|---|---|
| SD1.5 | IP-Adapter (Ye et al., 2023) | 45.65 | 57.82 | 54.58 |
| | IP-Adapter + OSPA | **53.89** | **60.59** | **60.51** |
| | IP-AdapterPlus (Ye et al., 2023) | 56.71 | 54.45 | 64.16 |
| | IP-AdapterPlus + OSPA | **59.90** | **57.43** | **67.68** |
| SDXL | InstantID (Wang et al., 2024) | 74.68 | 62.01 | 62.83 |
| | InstantID + OSPA | **75.89** | **63.13** | **63.96** |
| DiT | InfiniteYou (Jiang et al., 2025) | 57.48 | 51.74 | 53.33 |
| | InfiniteYou + OSPA | **60.47** | **54.38** | **56.48** |

where $\mathcal{A}_i = \frac{s_i - \text{mean}(\mathbf{s})}{\text{std}(\mathbf{s})}$ is the self-reward coefficient, $H$ is the shifted timestep sampling strategy proposed in SD3 (Esser et al., 2024), $\epsilon_\theta$ is the trainable policy model that is optimized to align its outputs with preferences, $\epsilon_{ref}$ is a fixed copy of the initial baseline model, which serves as a regularizer to prevent the policy model from deviating too far from the original distribution.

**Online Curriculum Preference Learning.** Although combining the self-preference sample generation and self-reward preference optimization module, we can align the policy model with the preference of the initial frozen sample generation model. However, as the policy model's preferences are continually optimized, our training process converges to a stable preference state. This is because the current model has largely exhausted its exploration within the constraints of the existing policy and exhibits significantly stronger preference alignment than the policy model itself. Therefore, offline updates confine the current model's preference optimization to a limited space. Consequently, we propose updating the frozen sample generation model with the optimized policy model after the optimization step, as shown in Fig. 3 (c). This design compels the model to continuously explore better alignments in an online update manner, thereby yielding more efficient self-preference. The complete procedure of our method is presented in the Algorithm 1.

## 4 EXPERIMENTS

### 4.1 EXPERIMENT SETUP

**Datasets.** We carefully curated the training data through a comprehensive screening procedure to train our OSPA model. Specifically, we utilize YOLOv8-Face [1] and insightface [2] technology to filter out high-quality data from the FaceCaption-15M (Dai et al., 2024) dataset. We constructed a dataset consisting of approximately 40,000 high-quality face images with size larger than $512 \times 512$ pixels. In addition, we leverage GPT-4o [3] to filter the face-related caption from JourneyDB (Sun et al., 2023), and then curate approximately 27,000 high-quality prompts for ID preservation generation. For evaluation, we carefully selected 30 reference face images from FFHQ (Karras et al., 2019), and 40 prompts from FastComposer (Xiao et al., 2025). For each reference image, we obtain 4 corresponding generated images for all prompts, resulting in a total of 4,800 identity-preserving images to compare the performance of different models.

**Evaluation Metrics.** To assess ID preservation preferences, we follow (Li et al., 2024b) and report the face similarity score, and CLIP-I score. For identity-preserving generation, we also apply the FLIP (Li et al., 2024a; Wang et al., 2025), a face-specific vision-language model built upon CLIP, for calculation of FLIP-I score to further evaluate the ID preservation of the generated images.

**Implementation Details.** We apply our plug-and-play framework to four representative models. Specifically, **for IP-Adapter** and **IP-Adapter-plus**, the model weights are initialized using pre-trained weights IP-Adapter-FaceID (Ye et al., 2023), only the adapter and projector weights are

---

[1] https://github.com/Yusepp/YOLOv8-Face

[2] https://github.com/deepinsight/insightface

[3] https://openai.com/index/gpt-4o-mini-advancing-cost-efficient-intelligence/

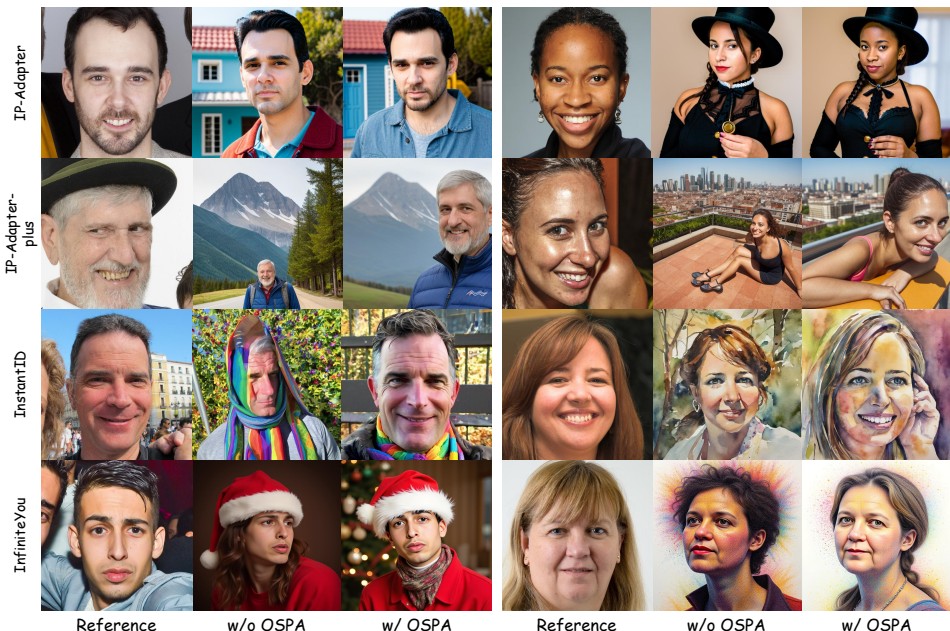

Figure 4: Qualitative comparison of preference generation results before and after optimization using our OSPA method in state-of-the-art baselines.

updated during training. For each generated data, 5 timesteps will be randomly sampled at a time for gradient update, and this step will be repeated 3 times. The learning rate is set to $1 \times 10^{-6}$ for a total of 2560 training steps. **For InstantID**, we train the adapter and projector weights. The learning rate is set to $1 \times 10^{-6}$ for a total of 2560 training steps. Each step generate 8 images and 7 timesteps will be randomly sampled at one time for gradient update. **For InfiniteYou**, we train the controlnet and projector weights. The remaining configurations are identical to those of InstantID. More details for all models will be provided in the supplementary material.

## 4.2 MAIN RESULTS

**Quantitative Comparison.** A quantitative analysis comparing our approach with baseline identity-preserving generation methods is summarized in Tab. 1, with results reported across various evaluation metrics. Quantitative evaluations show that our method improves identity preservation across all baseline models, achieving consistent and significant gains. Specifically, with IP-Adapter, IP-Adapter-plus, and InfiniteYou, our approach leads to a significant increase in face similarity, with gains of 8%, 3%, and 3%, respectively. Furthermore, our higher(at least a 3% improvement) CLIP-I and FLIP-I scores reflect enhanced overall subject consistency. Even for the strong InstantID baseline, our method achieves further gains across all metrics, with at least 1% improvement in face similarity, CLIP-I, and FLIP-I, demonstrating broad applicability and strong generalizability to identity-preserving models.

To further validate the effectiveness of our proposed framework, we also compared it against three existing training frameworks, as shown in the Tab. 2. Specifically, we first train the baseline using the supervised fine-tuning (SFT) framework on the highest-quality images generated by the baseline itself; however, its results did not surpass those of our method. More importantly, after training with SFT, its performance is slightly lower than that of the baseline itself. This indicates that the qual-

Table 2: Quantitative comparison with existing training frameworks, including SFT, ReFL(Chen et al., 2024), and DPO(Wallace et al., 2024), where IP-Adapter (Ye et al., 2023) as baseline model.

| Methods | Sim.↑ | CLIP-I↑ | FLIP-I↑ |
|---------|-------|---------|---------|
| baseline | 45.65 | 57.82 | 54.58 |
| + SFT | 42.60 | 56.69 | 54.24 |
| + ReFL | 51.32 | **60.90** | 59.93 |
| + DPO | 50.29 | 59.36 | 57.52 |
| **+ OSPA** | **53.89** | 60.59 | **60.51** |

ity of the generated images by the baseline is insufficient to meet the high-quality data requirements of the SFT training framework. However, our OSPA method leverages the preferences among the images generated by the baseline to steer its training toward aligning with its own preferences. Secondly, we compare our method with ID-Aligner (Chen et al., 2024), which enhances identity-preserving text-to-image generation with reward feedback learning. The baseline is trained using the reward feedback learning (ReFL) framework from ID-Aligner. The results in the Tab. 2 show that our method significantly outperforms the ReFL method across most metrics. It's worth noting that the ReFL method first requires training an additional reward models to provide reward signals, and encounter inefficient optimization caused by the offline feedback with the pre-collected preference datasets. Finally, we compare the baseline trained using direct preference optimization (DPO) with our method. As shown in the Tab. 2, our method consistently outperforms the DPO method. The main reasons is that we utilize the group-wise optimization approach that directly leverages reward scores, eliminating the need for pairwise preferences, and stably guide the model towards its own preferences during training.

**Qualitative Comparison.** The qualitative results are shown in Fig. 4. IP-Adapter demonstrates unsatisfied results, often altering skin tone (Column 5), whereas our OSPA effectively preserves skin tone in the generated images. Although IP-AdapterPlus embeds characters into scenes more effectively, it still shows noticeable shortcomings in identity preservation (Columns 2 and 5). By contrast, our method not only maintains facial similarity but also achieves natural blending with the background (Columns 3 and 6). Similarly, InstantID struggles to preserve facial identity (Columns 2 and 5), while our approach consistently retains identity fidelity (Columns 3 and 6). Overall, our proposed OSPA enhances both high-frequency details (e.g., color and lighting) and low-frequency structures, leading to consistently higher human preference scores.

**User Study.** We conduct a user study to compare OSPA with its baseline. A total of 21 participants evaluated 60 sample pairs, each assessing the facial similarity between generated images and the corresponding reference image. For each pair, participants indicated whether the baseline or OSPA result was more similar to the reference, with a "Tie" option if no clear difference was perceived. As shown in Fig. 5, the voting results clearly favor our method over the baselines.

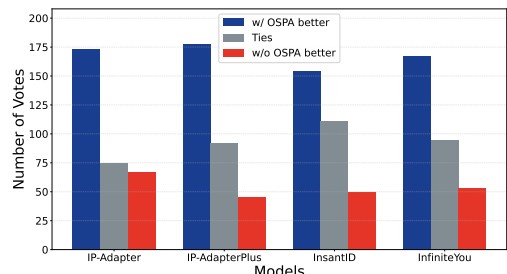

Figure 5: User study analysis.

## 4.3 ABLATION STUDIES

**Impact of Self-Preference Sample Generation.** We now investigate the effect of applying perturbation to face embeddings. Results in Tab. 3 show that perturbation yields consistent gains, highlighting its contribution to overall performance. Instead of relying solely on subtle differences between naturally produced samples, the perturbation introduces controlled identity degradation, making preference feedback clearer and more reliable. This enhanced supervision signal encourages the model to better recognize and penalize identity drift, ultimately strengthening identity consistency in the generated outputs.

Table 3: Quantitative comparison with and without the Self-Preference Sample Generation (SPSG) module, where IP-Adapter (Ye et al., 2023) as baseline model.

| Methods | Sim.↑ | CLIP-I↑ | FLIP-I↑ |
| --- | --- | --- | --- |
| baseline | 45.65 | 57.82 | 54.58 |
| + OSPA (w/o SPSG) | 52.32 | 60.55 | 60.10 |
| + OSPA (w/ SPSG) | **53.89** | **60.59** | **60.51** |

**Impact of Self-Reward Preference Optimization.** To assess the effectiveness of our proposed self-reward model, we compare OSPA using the self-reward mechanism against OSPA using the external Antelopev2 model (Deng et al., 2019). As shown in Tab. 4, the self-reward model yields consistent gains: +1.3% in identity preservation, +0.3% in CLIP-I, and +0.9% in FLIP-I. These improvements arise because the self-

Table 4: Quantitative comparison with and without the Self-Reward Preference Optimization module, where IP-Adapter (Ye et al., 2023) as baseline model.

| Methods | Sim.↑ | CLIP-I↑ | FLIP-I↑ |
| --- | --- | --- | --- |
| baseline | 45.65 | 57.82 | 54.58 |
| + OSPA (external) | 53.19 | 60.41 | 59.97 |
| + OSPA (self) | **53.89** | **60.59** | **60.51** |

reward model is derived directly from the identity-preserving text-to-image framework itself, enabling more tailored guidance and reducing the risk of suboptimal solutions that can occur when external reward models introduce distribution mismatches.

**Impact of Online Curriculum Preference Learning.** We conduct an ablation study to evaluate the role of online curriculum preference learning. As shown in Tab. 5, the online update improves face similarity from 51.13 to 53.89, demonstrating a clear gain in identity preservation. In contrast, offline updates cause notable declines in both identity fidelity and

Table 5: Quantitative comparison with and without the Online Curriculum Preference Learning method, where IP-Adapter (Ye et al., 2023) as baseline model.

| Methods | Sim.↑ | CLIP-I↑ | FLIP-I↑ |
|---|---|---|---|
| baseline | 45.65 | 57.82 | 54.58 |
| + OSPA (offline) | 51.13 | 60.03 | 59.58 |
| + OSPA (online) | **53.89** | **60.59** | **60.51** |

generation diversity, leading to suboptimal results. The advantage of the online strategy lies in real-time self-learning, which continuously enriches preference data and broadens the optimization space, resulting in more stable and effective identity-preserving generation.

## 4.4 FURTHER ANALYSIS

**Analysis of Perturbation Region and Strategy.** To investigate the impact of our proposed perturbation method on different regions of the generated image, we first generate a set of preference images using different perturbation intensities (noise intensity coefficient $\alpha$), and then calculate the cosine similarity of only face region, only background region, and prompt between the generated images, respectively. Fig. 6 reveals a clear decline in the cosine similarity for the face region between generated and reference images, while the similarities for both the background region (comparing perturbed and non-perturbed outputs) and the text prompt (against the generated images) remain stable. This suggests that our proposed perturbation strategy is sensitive to

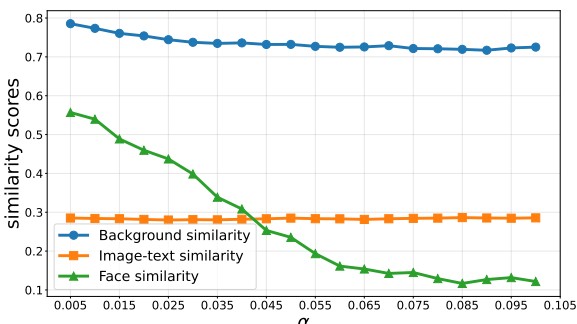

Figure 6: The sensitivity analysis of perturbation region by adding Gaussian noise to the image embedding. The similarity scores are calculated for the reference images with or without Gaussian noise perturbation. Results show that the embedding perturbation affects the face identity region more strongly, while having less impact on other regions.

face region, yet remains insensitive to background region and text similarity. This property of perturbing only the identity region while keeping other attributes invariant can guide the self-preference sample generation module to generating preference/unpreference groups, further guide the training of self-reward preference optimization.

To further investigate the impact of different perturbation strategies, we compared the visualization results of different noise types under different perturbation methods. As shown in Fig. 7 (a) and (b), directly adding Salt-and-Pepper or Gaussian noise to the pixels of reference images does not produce meaningful identity perturbation, making it ineffective for constructing the preference pairs. In addition, perturbing the embeddings using the Salt-and-Pepper noise severely degrades generation quality and erases identity information, as shown in Fig. 7 (c). In contrast, we perturbs face embeddings with Gaussian noise not only does not destroy the generated face images, but also maintains a certain difference from the original ID, making it well-suited for constructing preference data pairs, as shown in (Fig. 7 (d)). For more analysis of perturbation strategies, please refer to Appendix C

**Analysis of Hyperparameter Sensitivity.** We conduct sensitivity analysis on noise intensity coefficient $\alpha$ corresponding to Eq. 2 for our OSPA based on the baselines of IP-Adapter (Ye et al., 2023) and IP-AdapterPlus (Ye et al., 2023). It's worth noting that the image encoder within IP-Adapter is specifically designed for faces and is highly sensitive to facial images, while the image encoder in IP-AdapterPlus was not specifically trained on a face dataset, and therefore has weaker sensitivity to facial images. Therefore, based on the image encoder's sensitivity to faces, we choose a smaller $\alpha$ for a more sensitive baseline and a larger $\alpha$ for a less sensitive baseline. As shown in the Fig. 8 (a),

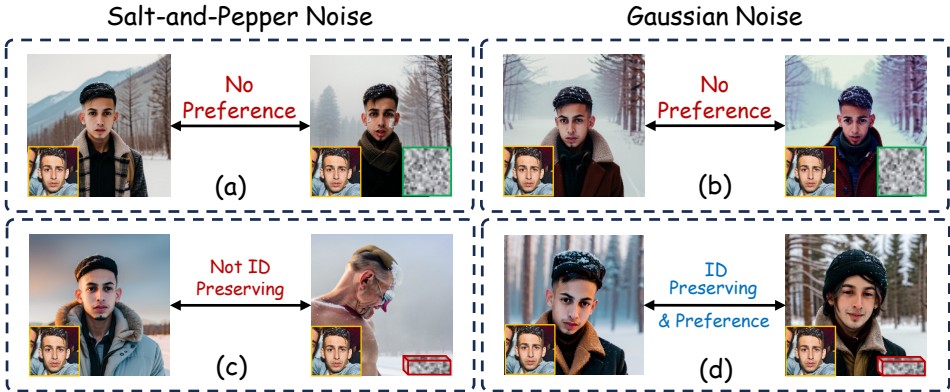

Figure 7: Qualitative comparisons with different self-preference sample generation strategies. Perturb the reference image with the salty noise image (a) and the Gaussian noise image (b); Perturb the reference image embedding with the salty noise (c) and Gaussian noise (d).

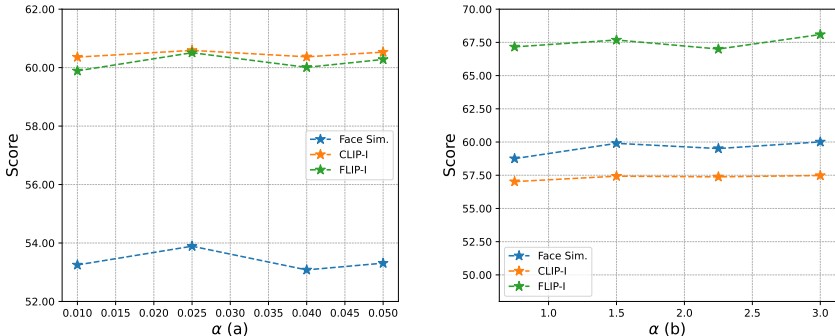

Figure 8: Sensitivity analysis on $\alpha$ for IP-Adapter (a) and IP-AdapterPlus (b) enhancing with our OSPA. Face similarity, CLIP-I, and FLIP-I are reported.

the all similarity scores remains stable as $\alpha$ varies from 0.01 to 0.05 for IP-Adapter enhancing with our OSPA method. In Fig. 8 (b), varying $\alpha$ between 0.75 and 3.00 has minimal effect. Both these results confirm the stability of the self-perturbation strategy using the noise intensity coefficient $\alpha$. For more sensitivity analyses, please refer to Appendix C.

For more discussion, analysis, additional applications, failure cases, and visualization results, please refer to Appendix C and D.

## 5 CONCLUSION

In this work, we revisit identity-preserving text-to-image generation and highlight the often-overlooked importance of diversity in achieving reliable identity fidelity. We propose Online Self-Preference Alignment (OSPA), a simple yet effective plug-and-play framework that eliminates the need for external preference datasets or reward models. Instead, OSPA leverages self-generated preference data, intrinsic self-reward mechanisms, and an online update strategy to broaden the optimization space. Extensive experiments across multiple state-of-the-art baselines demonstrate that OSPA consistently enhances identity preservation while maintaining visual quality, establishing it as a practical and generalizable solution for identity-preserving generation.

**Limitations.** Online Self-Preference Alignment requires pre-trained identity-preserving text-to-image generation models, resulting in the degree of improvement in identity quality that depends largely on the baseline model. To further enhance performance, promising directions include scaling up the model architecture and improving the preference optimization method. While OSPA demonstrates strong generalizability across different face-oriented models, its performance may be contingent on the characteristics of the base model and data, and a systematic analysis of these influencing factors is needed for reliable cross-domain application.

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

# APPENDIX

## A PRELIMINARY

**Diffusion Models.** Diffusion models are a class of generative models capable of synthesizing desired data samples through iterative denoising. A conventional diffusion training encapsulates two procedures, the forward diffusion process and the reverse denoising process. During the diffusion process, noise $\epsilon$ is sampled and added to the data sample $x_0$ based on a predefined noise schedule. This process yields a noisy sample $x_t$ at timestep $t$. Conversely, during the denoising process, a denoising model $\epsilon_\theta$ takes $x_t$, $t$, and optional additional conditions $C$ as inputs to predict the added noise. The optimization process can be articulated as:

$$\mathcal{L}_{\text{diff}} = \mathbb{E}_{x_0,\epsilon,t}(\|\epsilon - \epsilon_\theta(x_t, t, C)\|) \tag{7}$$

The denoising model $\epsilon_\theta$ is predominantly a UNet composed of residual blocks, self-attention layers, and cross-attention layers.

**Reward Feedback Learning.** Reward feedback learning is an effective preference learning method that utilizes a well-trained human preference reward model $R$ for preference fine-tuning. It begins with an input prompt $p$, initializing a latent variable $x_T$ at random. The latent variable is then progressively denoised until reaching a randomly selected timestep $t$. At this point, the denoised image $x_0'$ is directly predicted from $x_t$. The reward model $R$ is then applied to this denoised image, generating the expected preference score $R_\theta(c, x_0')$. ReFL maximizes such preference scores to align the generated images more closely with human preferences:

$$\mathcal{L}_{\text{ReFL}}(\theta) = \mathbb{E}_{c \sim p(c)} \mathbb{E}_{x_0' \sim p(x_0'|c)}(-R(x_0', c)) \tag{8}$$

## B MORE IMPLEMENTATION DETAILS

**Similarity Calculation Details.** In Eq. 5, sim $(\cdot)$ denotes the computation of cosine similarity between two image embeddings, see Eq. 9. For facial embeddings, calculating cosine similarity places greater emphasis on the directional alignment of features, providing a more intrinsic measure of identity patterns. For background similarity, image-text similarity, face similarity in Fig. 6, we also calculate the score using cosine similarity. Specially, we extract CLIP feature to calculate cosine similarity for background similarity and image-text similarity, and extract face embedding to calculate cosine similarity for face similarity.

$$\text{sim}(\boldsymbol{v}_{\text{ref}}, \boldsymbol{v}_{\text{gen}}^i) = \frac{\boldsymbol{v}_{\text{ref}} \cdot \boldsymbol{v}_{\text{gen}}^i}{\|\boldsymbol{v}_{\text{ref}}\| \|\boldsymbol{v}_{\text{gen}}^i\|} \tag{9}$$

**Datasets Details.** Tab. 6 shows all the prompts used for identity-preserving text-to-image generation. Additionally, we have compared the evaluation set sizes used by different methods with ours, as detailed in Tab. 7. Our evaluation set maintains a comparable scale to those of other approaches, and the reference images encompass diverse races, ages, and genders, ensuring broad coverage and rationality of our benchmark. Regarding of caption filtering by GPT-4o, we use the prompt "Please analyze whether the given sentence describes a single human (we do not want descriptions of multiple humans or sentences that do not describe any human at all). And please do not include images depicting the back. If it does, respond with 'yes'. If it does not, respond with 'no'." to filter the caption.

**Training Details with OSPA.** In Eq. 6, $H(1, T)$ denotes the temporal shift function $t_m = \frac{\beta t_n}{1+(\beta-1)t_n}$, $t_m \sim \mathcal{U}(1, T)$. In practice, we set the shift value $\beta$ to 1. For all baseline experiments, we adhered to the original scheduler and its default configurations for sampling. Regarding of training IP-Adapter with OSPA, For each reference image, we randomly select a prompt and generate 16 images. Among these, 12 images are perturbed by adding noise. The number of inference steps during sampling is set to 30, with a guidance scale of 4. After each sampling iteration, the model is updated 15 times using randomly selected time steps. Noise is added to the extracted facial features with an alpha value of 0.025, and similarity is compared after passing through the image projector. Regarding of training IP-Adapter Plus with OSPA, The adapter scale is set to 0.3, Noise is added to

Table 6: Evaluation prompts.

| Index | Prompts |
|---|---|
| 1 | "a painting of a person in the style of Banksy" |
| 2 | "a painting of a person in the style of Vincent Van Gogh" |
| 3 | "a colorful graffiti painting of a person" |
| 4 | "a watercolor painting of a person" |
| 5 | "a Greek marble sculpture of a person" |
| 6 | "a street art mural of a person" |
| 7 | "a black and white photograph of a person" |
| 8 | "a pointillism painting of a person" |
| 9 | "a Japanese woodblock print of a person" |
| 10 | "a street art stencil of a person" |
| 11 | "a person wearing a red hat" |
| 12 | "a person wearing a santa hat" |
| 13 | "a person wearing a rainbow scarf" |
| 14 | "a person wearing a black top hat and a monocle" |
| 15 | "a person in a chef outfit" |
| 16 | "a person in a firefighter outfit" |
| 17 | "a person in a police outfit" |
| 18 | "a person wearing pink glasses" |
| 19 | "a person wearing a yellow shirt" |
| 20 | "a person in a purple wizard outfit" |
| 21 | "a person riding a horse" |
| 22 | "a person holding a glass of wine" |
| 23 | "a person holding a piece of cake" |
| 24 | "a person giving a lecture" |
| 25 | "a person reading a book" |
| 26 | "a person gardening in the backyard" |
| 27 | "a person cooking a meal" |
| 28 | "a person working out at the gym" |
| 29 | "a person walking the dog" |
| 30 | "a person baking cookies" |
| 31 | "a person in the jungle" |
| 32 | "a person in the snow" |
| 33 | "a person on the beach" |
| 34 | "a person on a cobblestone street" |
| 35 | "a person on top of pink fabric" |
| 36 | "a person on top of a wooden floor" |
| 37 | "a person with a city in the background" |
| 38 | "a person with a mountain in the background" |
| 39 | "a person with a blue house in the background" |
| 40 | "a person on top of a purple rug in a forest" |

Table 7: Comparison of evaluation dataset of different methods.

| Methods | Reference | Prompt | Sample | Total Image |
|---|---|---|---|---|
| ID-Aligner | 20 | 40 | **5** | 4000 |
| InfiniteYou | 15 | **200** | - | 1497 |
| **Ours** | **30** | 40 | 4 | **4800** |

the extracted facial features after passing through image encoder with an alpha value of 1.5, while all other configurations remain the same as those of IP-Adapter. Regarding of training InstantID with OSPA, For each reference image, we randomly select a prompt and generate 8 images. Among these, 4 images are perturbed by adding noise. The number of inference steps during sampling is set to 30, with a ControlNet conditioning scale of 0.2, an adapter scale of 0.8, and a guidance scale of

5. After each sampling iteration, the model is updated 7 times using randomly selected time steps. Noise is added to the extracted facial features with an alpha value of 0.04, and similarity is compared after passing through the image projector. Regarding of training InfiniteYou with OSPA, For each reference image, we randomly select a prompt and generate 8 images. Among these, 4 images are perturbed by adding noise. The number of inference steps during sampling is set to 10, with a ControlNet conditioning scale of 0.8. Noise is added to the extracted facial features with an alpha value of 0.04, and similarity is compared after passing through the image projector.

**Training Details with other Framework.** We conducted comparative experiments on IP-Adapter using different training frameworks. Regarding of SFT training of IP-Adapter, we employed the Best-of-N method (N=16) to select the generated image with the highest similarity score for supervised fine-tuning. To ensure a fair comparison, we used the same number of training steps as in OSPA, with the batch size set to 16—matching the number of images sampled per step under the OSPA framework for IP-Adapter. All other training parameters remained consistent with those in the original IP-Adapter configuration. Regarding of DPO(Wallace et al., 2024) training of IP-Adapter, we allocated the same total training time as in OSPA, resulting in approximately 16 times more training steps. The regularization coefficient was set to 8000, while all other configurations aligned with the IP-Adapter setup under the OSPA framework. Regarding of ReFL(Chen et al., 2024) training of IP-Adapter, the number of training steps was kept consistent with that of IP-Adapter under the OSPA framework. Other experimental parameters followed the ID-Aligner settings, with one key modification: in practice, the face detector used in ID-Aligner failed to detect faces in most generated images. Therefore, we replaced it with RetinaFace to ensure reliable face detection. In SFT and DPO methods, the specific generative parameters used during sampling were identical to those applied in the IP-Adapter training under the OSPA framework.

## C  ADDITIONAL DISCUSSION AND ANALYSIS

**More Sensitivity Analysis.** To explore how to adaptively select the hyperparameter $\alpha$ based on the sensitivity of a baseline image encoder to face images, we assess the difference in face similarity between the noise-perturbed generation image and the reference image in both small and large $\alpha$ value ranges. As shown in Fig. 9, for image encoders with high face sensitivity, the face similarity between the generated image—perturbed within small $\alpha$ ranges—and the reference face exhibits a pronounced decline. In contrast, encoders with low face sensitivity maintain consistently high similarity scores. In addition, when $\alpha$ varies over a larger range, the similarity between the generated image (perturbed using noise) and the reference image remains stably low for the high-sensitive encoder of face, whereas encoders with lower face sensitivity exhibit a noticeable decline in similarity. This comparison demonstrates that we can adaptively select an appropriate $\alpha$ value based on the sensitivity of the image encoder to the face in the baseline model. Furthermore, please refer to Fig. 8 and Fig. 10 for more the robustness analysis of the hyperparameter $\alpha$.

**More Perturbation Strategy Analysis**. We here explore more perturbation strategies for constructing preferred and unpreferred face images, including randomly swapping values within the face embedding and randomly replacing parts of the face embedding with Gaussian noise. As shown in Fig. 11 (a), under the same swapping manner, the generated identity exhibit significant differences, ranging from images highly similar to the original identity to those markedly different from it. As shown in Fig. 11 (b), the generated images obtained using the perturbation of randomly replacing parts of the face embedding do not show significant differences in identity, making it impossible to construct effective preference data pairs to

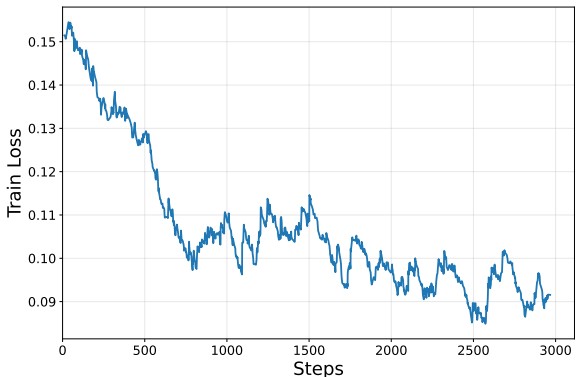

Figure 12: Training convergence analysis for training loss curve on IP-Adapter with our OSPA.

guide model training. In contrast, our perturbation method can construct the preference images with identity preservation by randomly adding Gaussian noise to all embeddings.

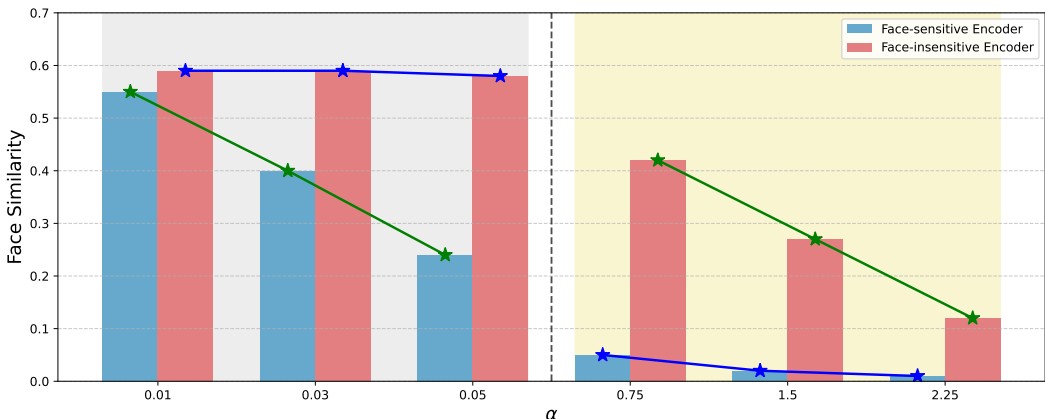

Figure 9: Analysis of adaptive noise adjustment strategy. We compare two types of image encoder in IP-Adapter and IP-AdapterPlus, which are face-sensitive encoder and face-insensitive encoder, within in both small and large $\alpha$ value ranges. The green line shows a significant downward trend, while the blue line area remains stable, indicating that different $\alpha$ values can be adaptively selected based on the sensitivity of image encoding to the face.

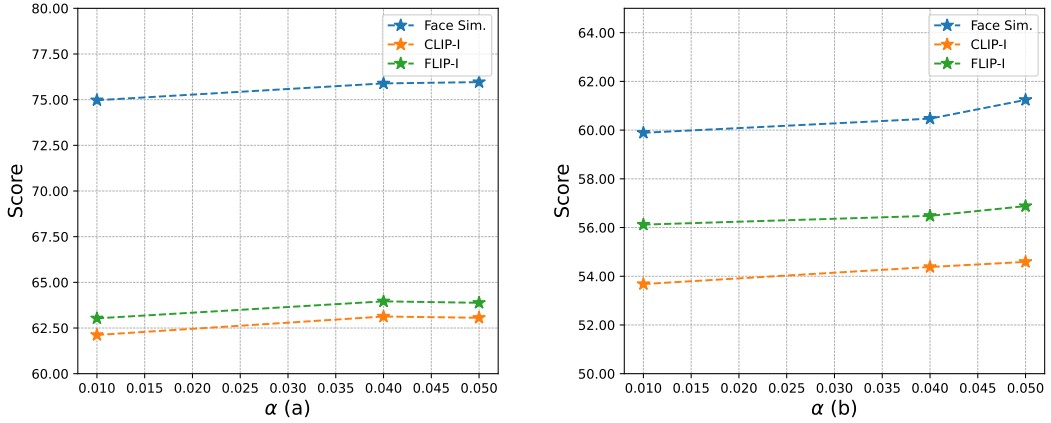

Figure 10: Sensitivity analysis on $\alpha$ for InstantID (a) and InfiniteYou (b) enhancing with our OSPA. Face similarity, CLIP-I, and FLIP-I are reported.

**Training Convergence Analysis.** To determine the convergence point and mitigate the risk of overfitting, we employed the training loss curve as a key indicator of training dynamics (see Fig. 12). Due to training time constraints, we stopped the training when the loss curve began to flatten slightly. Our model training has not yet reached a fully converged state. If training were to continue, the model's performance might further improve, though there could also be risks of over-exploration.

**Results on Challenging Reference Images.** In Fig. 13, we present the results of the IP-Adapter enhancing with our OSPA, which the reference images within extreme scenarios, such as reference images with blurry or glasses, extreme style changes (cartoon), and generating real-world photos from stylized images. This visualization demonstrates that the IP-Adapter enhancing with our OSPA effectively maintains identity fidelity even in extreme scenarios, indicating that our method has good generalization ability.

**Results on Complex Text Prompts.** Fig. 14 shows the results on complex text prompts. It can be observed that our method not only maintains superior consistency in identity preservation but also performs well in text-image alignment.

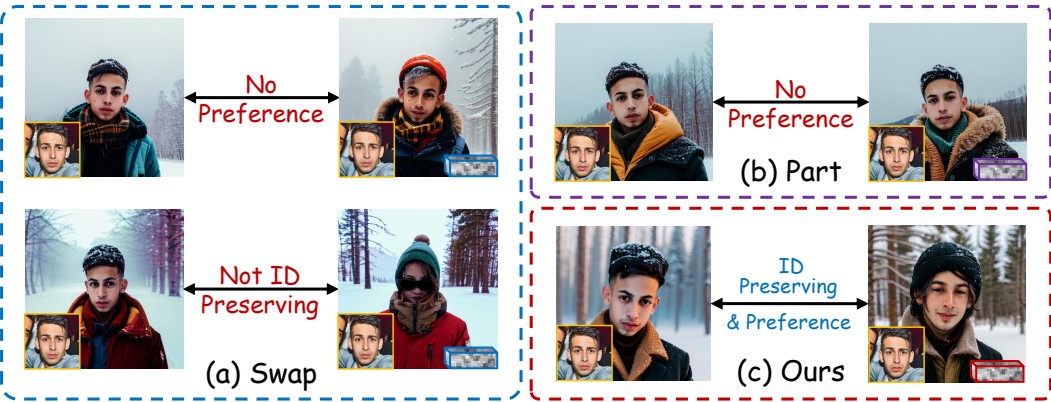

Figure 11: Qualitative comparisons of more perturbation strategies. (a) Randomly swapping embeddings; (b) Randomly replacing parts of the embeddings with Gaussian noise; (c) Randomly embed Gaussian noise into all embeddings.

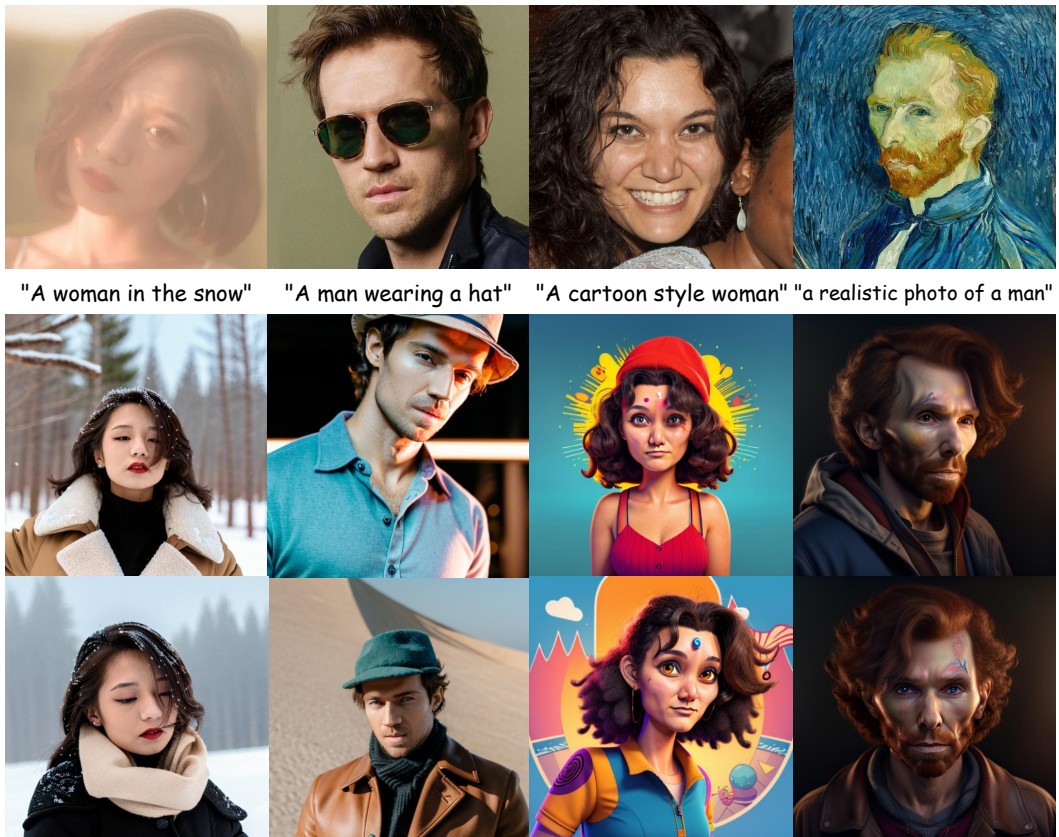

Figure 13: Visualization results on IP-Adapter enhancing with our OSPA method in some extreme scenarios, including reference images with blurry or glasses, extreme style changes (cartoon), and generating real-world photos from stylized images.

**Failure Cases.** We provide failure cases of the IP-Adapter enhancing with our OSPA in Fig. 15. Our current version of OSPA struggles to improve identity fidelity in masked, low-light, and stylized images. One possible reason is our framework is built upon the foundational fact that the base model possesses an effective understanding of facial attributes. Consequently, it struggles to extract effective facial features from reference images that suffer from significant information loss, such as

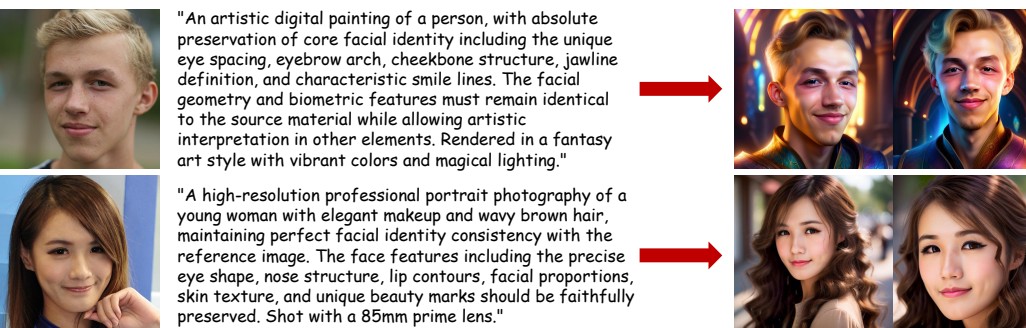

Figure 14: Visualization results with complex text prompts on IP-Adapter enhancing with our OSPA method.

those with large occluded areas or captured under low-light conditions. Moreover, since the baseline model was trained on datasets of real faces, it fails to generalize to highly stylized reference images. The substantial domain gap prevents the model from establishing a meaningful correspondence to the real facial domain, resulting in an inability to generate realistic human faces. Furthermore, since our method primarily focuses on face images, its identity fidelity performance in other domains still needs improvement, as shown in Fig. 16. One possible future direction is to utilize more extensive data for OSPA learning, thereby improving identity fidelity in various scenarios.

**Discussion and Future Work.** The current OSPA framework has only been validated for identity-preserving generation tasks. We will Explore the transferability of OSPA to a broader range of vision tasks beyond face generation, such as general image editing or artistic style transfer, represents a promising future direction. Moreover, within the online self-enhancing framework, no existing work has provided a comprehensive analysis of its convergence(Hamadanian et al., 2023)(Gupta et al., 2025)(Bai et al., 2025). This remains a challenging yet valuable research direction. We will rigorously prove the convergence of our OSPA framework in future work. Additionally, our approach has not yet fully explored the stability during continued training and the potential risks of over-exploration. We will systematically investigate these aspects in our subsequent work, including evaluating long-term performance degradation and developing strategies to mitigate excessive exploration.

# D  ADDITIONAL QUALITATIVE RESULTS

**Results of Image-Text Alignment.** Fig. 17 shows the image-text alignment. From this figure, we can see that our method maintains high-quality ID generation without damaging the text-image alignment capability, and can even implicitly improve the model's text-image alignment capability.

**More Visual Comparison Results.** We provide more visualization results of our method and baselines, including IP-Adapter (see Fig. 18), IP-AdapterPlus (see Fig. 19), InstantID (see Fig. 20), and InfiniteYou (The original image size is 1152x864) (see Fig. 21).

# E  THE USE OF LLMS

We used LLM to check the grammatical errors of some sentences and the usage of vocabulary in some sentences. And some of the sentences referred to the suggestions given by LLM.

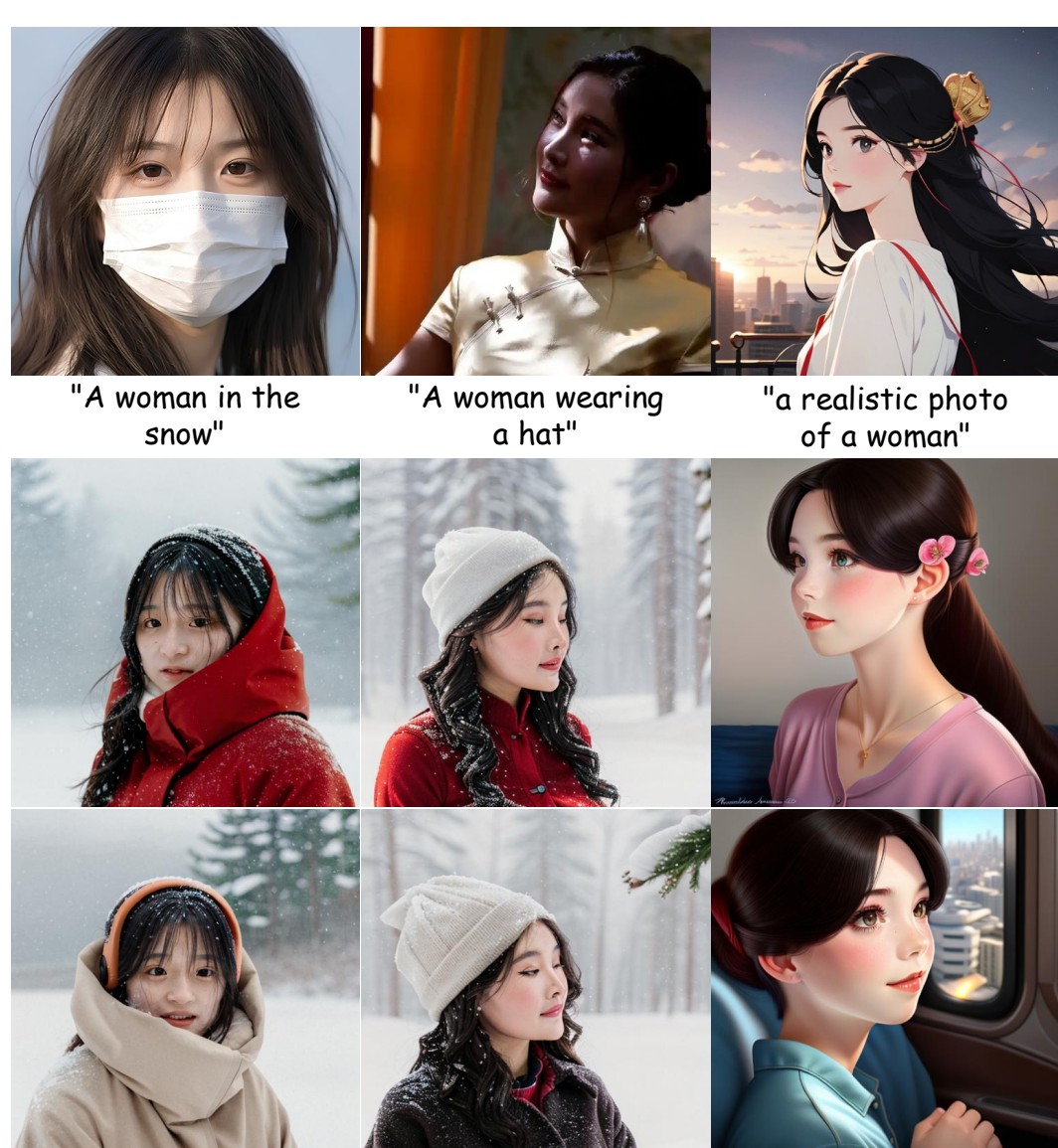

Figure 15: Visualization of failure cases of enhancing IP-Adapter with our OSPA on reference images containing large occlusions (e.g., masks), low-light conditions, or extreme stylization.

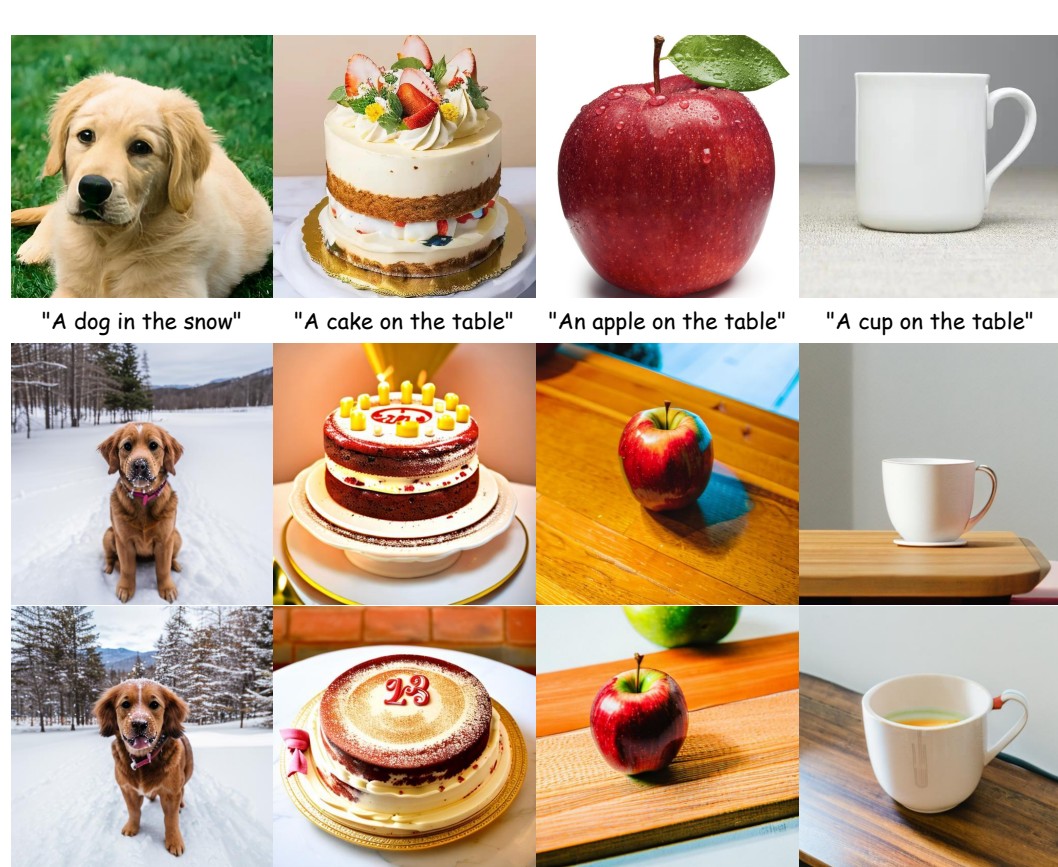

"A dog in the snow"    "A cake on the table"    "An apple on the table"    "A cup on the table"

Figure 16: Visualization of failure cases on reference images from other domains when enhancing IP-Adapter with our OSPA method.

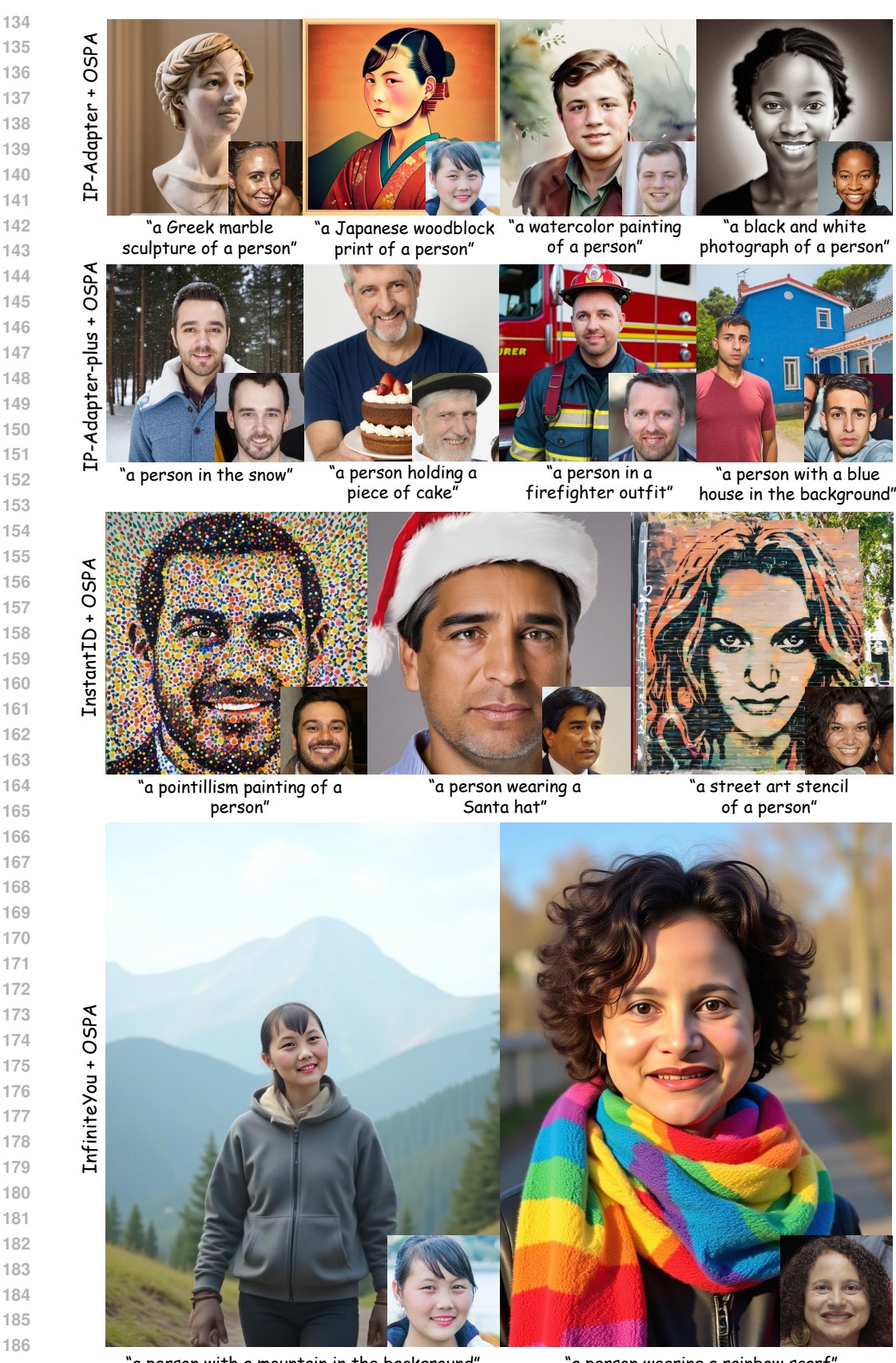

Figure 17: OSPA generates identity-preserved images with exceptional text-image alignment.

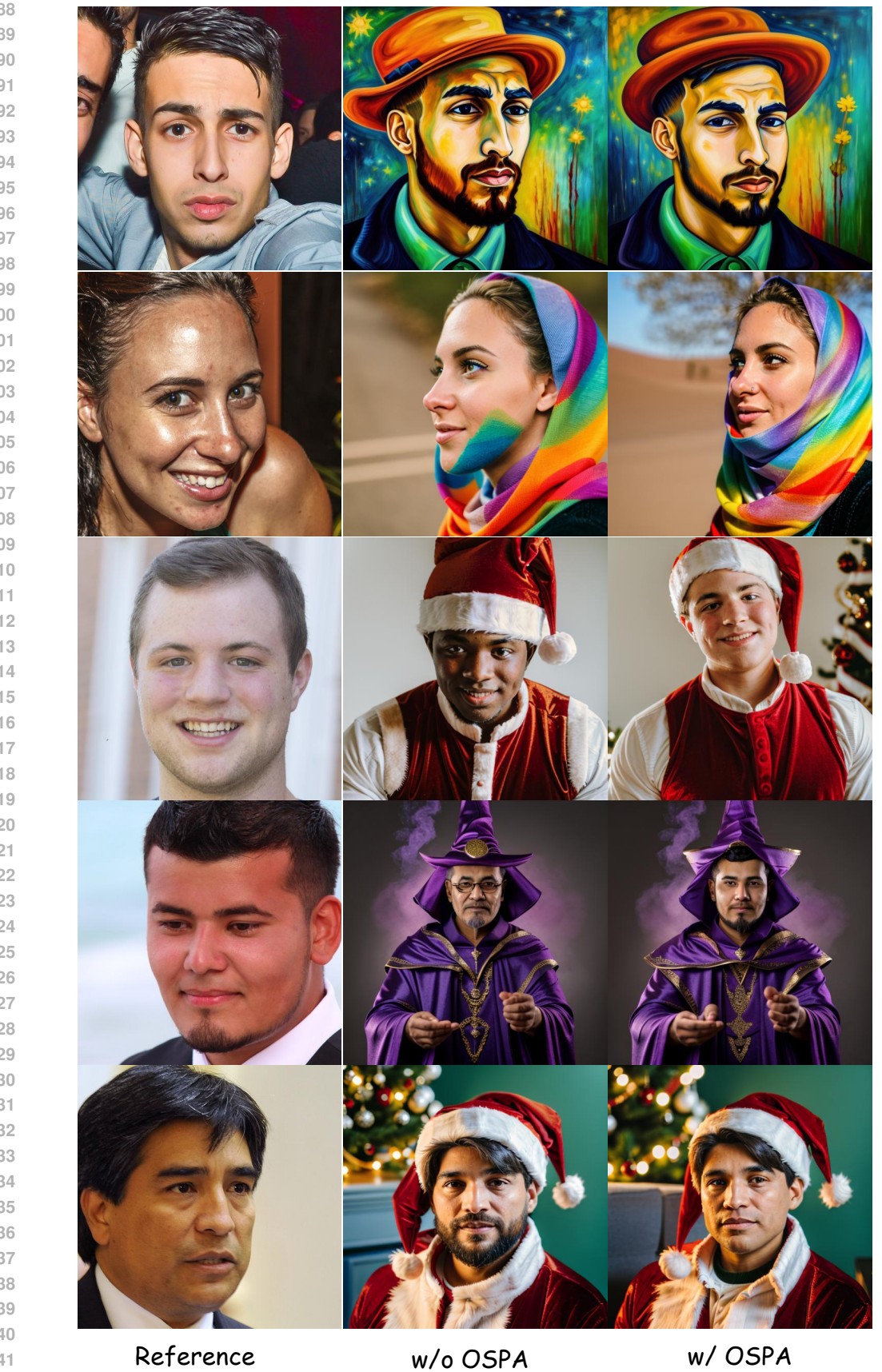

Reference          w/o OSPA          w/ OSPA

Figure 18: Qualitative comparisons with IP-Adapter.

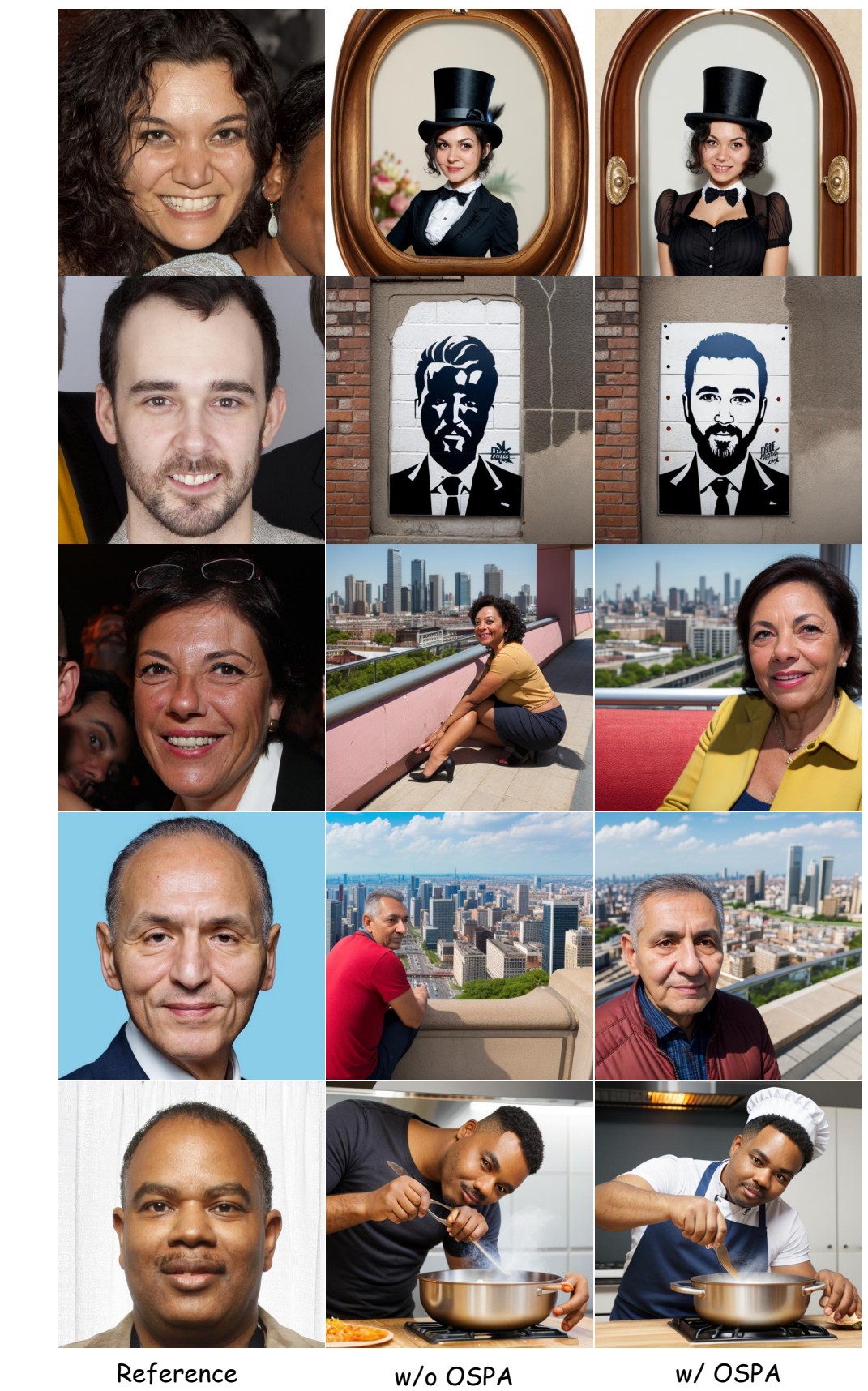

Reference       w/o OSPA       w/ OSPA

Figure 19: Qualitative comparisons with IP-AdapterPlus.

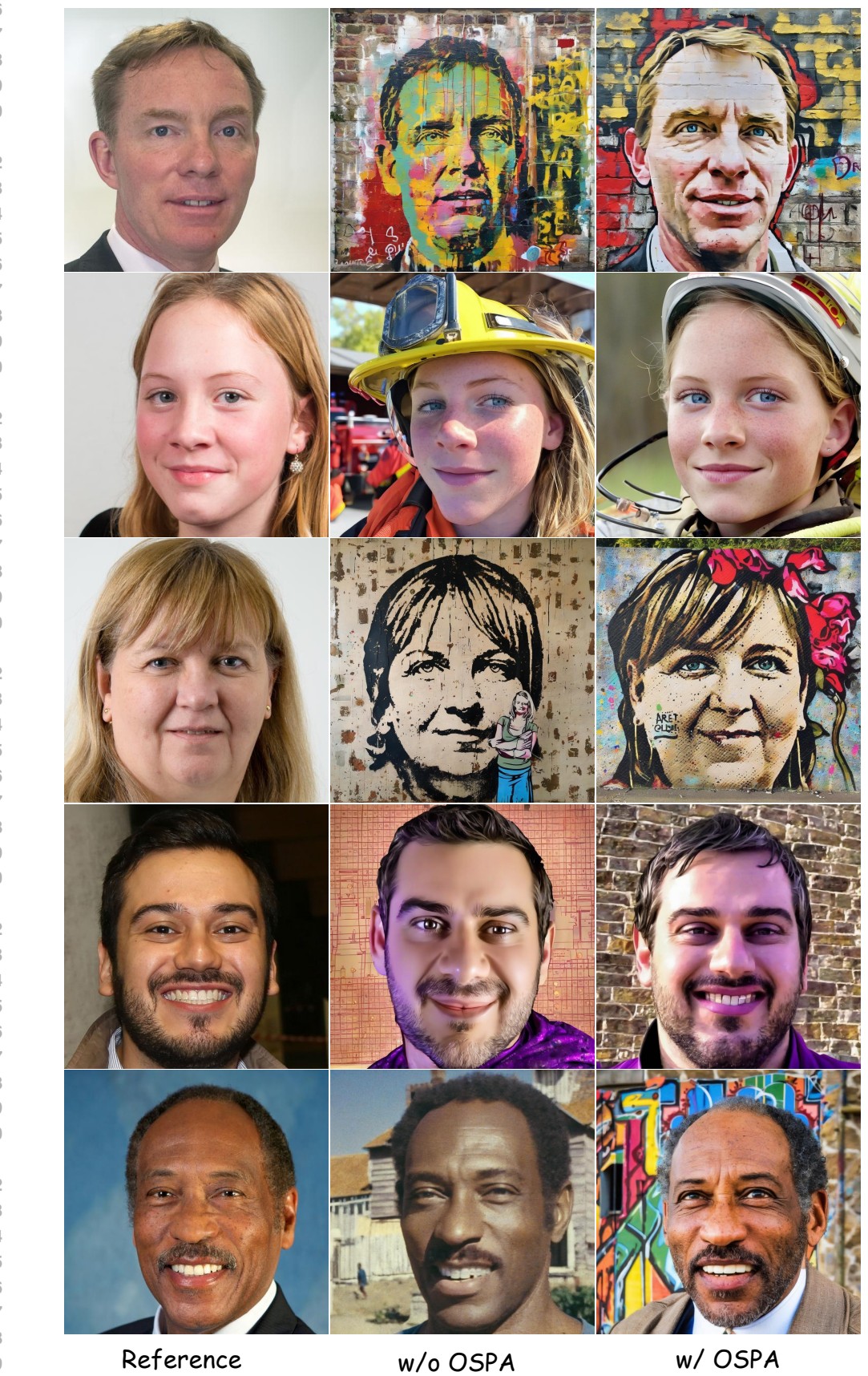

Reference        w/o OSPA        w/ OSPA

Figure 20: Qualitative comparisons with InstantID.

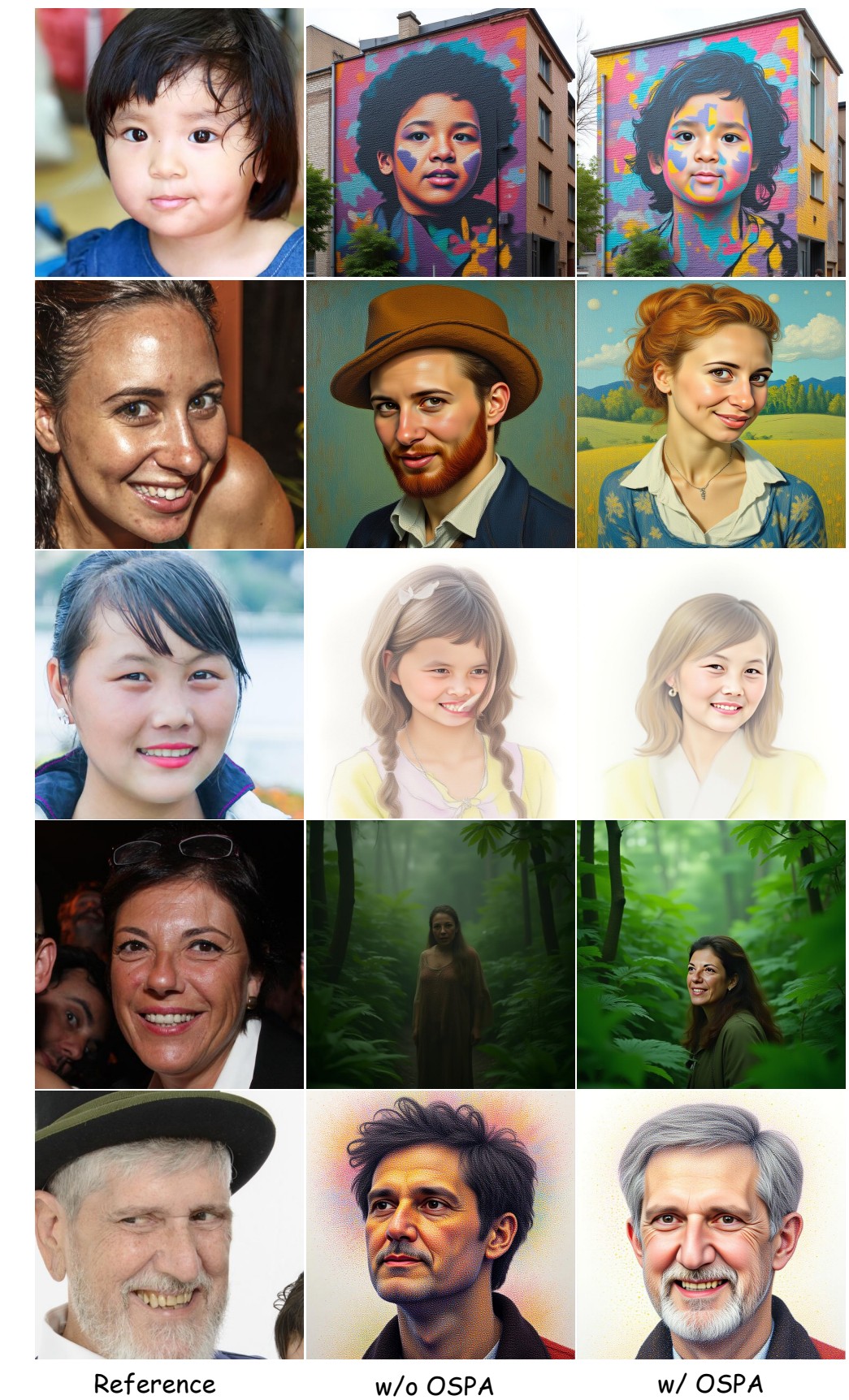

Reference      w/o OSPA      w/ OSPA

Figure 21: Qualitative comparisons with InfiniteYou.