# OpenReview forum: "OSPA: Enhancing Identity-Preserving Image Generation via Online Self-Preference Alignment"
_ICLR.cc/2026/Conference — Submitted to ICLR 2026_

### Official Review · Reviewer_kjRG · 2025-10-25

**Soundness:** 3
**Presentation:** 3
**Contribution:** 3
**Rating:** 6
**Confidence:** 4

**Summary:**

The paper presents Online Self-Preference Alignment (OSPA), a novel framework designed to improve identity-preserving text-to-image generation. The core motivation is to bypass the reliance on expensive human-annotated datasets or separately trained external reward models typically used in RLHF. OSPA achieves this by constructing a self-contained alignment loop: it generates its own preference pairs by perturbing identity embeddings, utilizes a frozen, pre-trained identity encoder to calculate "self-reward" scores based on cosine similarity, and employs an online curriculum learning strategy to iteratively refine the generative policy. Extensive experiments on four ID-preserving models (IP-Adapter, IP-AdapterPlus, InstantID, and InfiniteYou) demonstrate that OSPA consistently enhances identity fidelity while maintaining visual quality.

**Strengths:**

The most significant strength of this work lies in its pragmatic approach to a major bottleneck in personalized generation: the high cost of alignment. By ingeniously formulating a completely automated, self-referential optimization loop, OSPA effectively removes the need for external supervision during the alignment phase. This "plug-and-play" nature, verified across multiple diverse and strong baselines, highlights excellent generality and high potential for practical application in scalable model customization.

The quality of execution and clarity of presentation are also commendable. The paper is well-structured, and the core methodology is communicated effectively, particularly through well-designed figures like Figure 3. The empirical validation is robust, showing consistent quantitative improvements across various metrics (Face Sim, CLIP-I, FLIP-I) and baselines, suggesting that the proposed Group Preference Optimization (GPO) on self-generated data is a stable and effective training objective.

**Weaknesses:**

A primary conceptual weakness is the potentially misleading framing of "Self-Preference" and the unverified reliance on the specific pre-trained identity encoder. The "reward" signal is not truly intrinsic to the generative model but is distilled from a frozen, external discriminative model (the face encoder). The entire framework's upper bound is thus locked by this specific encoder's ability to act as a perfect proxy for human identity perception. The paper lacks a critical analysis of how sensitive the entire system is to the choice of this encoder. If the encoder has biases (e.g., focusing too much on low-level textures rather than high-level facial structure), OSPA will amplify these biases.

Furthermore, the experimental section lacks a crucial, simpler baseline to justify the complexity of the proposed preference optimization framework. Since the method relies entirely on the frozen ID encoder for scoring, a straightforward "Rejection Sampling Fine-tuning" approach—generating N samples, selecting the best one based on the same ID encoder score, and performing standard supervised fine-tuning—should be compared. If OSPA does not significantly outperform this much simpler strategy, the necessity of the complex paired-sample generation and GPO loss becomes questionable.

Finally, the method appears highly sensitive to the noise intensity hyperparameter α, as indicated in Figure 6, where performance degrades sharply outside a narrow window. The manuscript does not sufficiently detail the strategy for selecting α across the widely different baselines (e.g., IP-Adapter vs. InstantID). If fine-grained, per-model grid search is required for this parameter, it undermines the claimed "plug-and-play" ease of use and suggests potential fragility in new applications.

**Questions:**

Could you provide an ablation study or at least a discussion on replacing the currently used identity encoder with a different one (e.g., a different face recognition architecture, or a weaker generic encoder)? This is critical to verify whether the success of OSPA is due to the general framework or specifically tied to the high quality of the chosen "judge" encoder.

How does OSPA compare quantitatively to a simpler "Generate-Filter-Finetune" baseline using the same ID encoder as the filter? Demonstrating a clear margin over this baseline is necessary to robustly justify the added complexity of your online paired-preference learning framework.

Please clarify the exact strategy used for selecting the critical noise hyperparameter α for each of the four baselines. Was a single value universally effective, or was individual tuning necessary? If tuning was needed, how sensitive are the results to small variations in α for the different architectures?

---

> ### Author Response · Authors · 2025-11-27
> **Official Comment by Authors (Part 1)**
>
> Thank you for your kind approval (e.g., **well-structured manuscript**, **robust validation**, and **a stable and effective training objective**). Below, we provide detailed responses to your insightful comments.
>
> **Response to Weaknesses**
> ---
>
> **W1-(1). A primary conceptual weakness is the potentially misleading framing of "Self-Preference" and the unverified reliance on the specific pre-trained identity encoder. The "reward" signal is not truly intrinsic to the generative model but is distilled from a frozen, external discriminative model (the face encoder).**
> In our experiments, we did not rely on a single face encoder as the reward model. It is important to note that the ID-preserving generation model consists of three components: a face encoder, a feature adaptation module, and a U-Net. The face encoder serves as an internal core component of the model, responsible for extracting ID features and participating in the entire generation pipeline, rather than being an external independent model. Since the encoder is an internal module, our reward signals are generated by computing feature similarity between the generated image (produced through the aforementioned modules) and the reference image using this internal encoder. Therefore, our approach remains fundamentally "self-preference" in nature.
>
> **W1-(2). The entire framework's upper bound is thus locked by this specific encoder's ability to act as a perfect proxy for human identity perception.**
> The performance ceiling of our method is collectively determined by all components of the base model, including both the encoder and U-Net. The image generation quality produced by the UNet component serves as a critical factor influencing model training, while the encoder's evaluation of generated images provides essential guidance for the training process.
>
> **W1-(3). The paper lacks a critical analysis of how sensitive the entire system is to the choice of this encoder. If the encoder has biases (e.g., focusing too much on low-level textures rather than high-level facial structure), OSPA will amplify these biases.**
> For a specific ID generation model, if it inherently cannot handle particular biases (such as heavily occluded cases), then our OSPA will indeed fail to process such data. Nevertheless, our method remains effective in enhancing the model's capabilities under normal conditions.
>
> **W2. Furthermore, the experimental section lacks a crucial, simpler baseline to justify the complexity of the proposed preference optimization framework. Since the method relies entirely on the frozen ID encoder for scoring, a straightforward "Rejection Sampling Fine-tuning" approach—generating N samples, selecting the best one based on the same ID encoder score, and performing standard supervised fine-tuning—should be compared. If OSPA does not significantly outperform this much simpler strategy, the necessity of the complex paired-sample generation and GPO loss becomes questionable.**\
> We thank the reviewer for raising the question regarding the rationality of our framework's complexity. In strict accordance with your suggestion, we have conducted comparative experiments, and the results demonstrate that our method significantly outperforms the supervised fine-tuning method, thereby validating the rationality of our architectural design. More detailed analyses and experimental results are provided in our response to Q2.
>
> ##
>
> **W3. Finally, the method appears highly sensitive to the noise intensity hyperparameter $\alpha$, as indicated in Figure 6, where performance degrades sharply outside a narrow window. The manuscript does not sufficiently detail the strategy for selecting $\alpha$ across the widely different baselines (e.g., IP-Adapter vs. InstantID). If fine-grained, per-model grid search is required for this parameter, it undermines the claimed "plug-and-play" ease of use and suggests potential fragility in new applications.**\
> We thank the reviewer for the comment. Fig. 6 demonstrates that adding noise to the embedding space significantly affects the identity itself while having minimal impact on the background and image-text alignment. The sensitivity of hyperparameter $\alpha$ to face similarity is precisely what we require. It is important to note that the scores in Fig. 6, which vary with hyperparameter $\alpha$, do not represent the final performance but rather the impact of this perturbation method on the generated images. To further illustrate the effect of hyperparameter $\alpha$ on the final performance, we conducted a sensitivity analysis of $\alpha$, with detailed results presented i**n Fig. 8** and **10 in the revised version**. This table demonstrates the robustness of hyperparameter $\alpha$ to the final outcome.

---

> ### Author Response · Authors · 2025-11-27
> **Official Comment by Authors (Part 2)**
>
> **Response to Questions**
> ---
>
> **Q1. Could you provide an ablation study or at least a discussion on replacing the currently used identity encoder with a different one (e.g., a different face recognition architecture, or a weaker generic encoder)? This is critical to verify whether the success of OSPA is due to the general framework or specifically tied to the high quality of the chosen "judge" encoder.** \
> We have conducted experiments on four distinct baselines (i.e., four different encoders). The results demonstrate that our method consistently improves baseline performance across all encoders, confirming its robustness with diverse encoder architectures. When replacing the original encoder of the model (as shown **in Tab. 4 in the revised version**), performance improvements can still be observed, though they are less substantial compared to using the model's native encoder. This is because encoder replacement introduces misalignment with the original model configuration. These findings further validate that our self-preference approach is fundamentally sound.
>
> **Q2. How does OSPA compare quantitatively to a simpler "Generate-Filter-Finetune" baseline using the same ID encoder as the filter? Demonstrating a clear margin over this baseline is necessary to robustly justify the added complexity of your online paired-preference learning framework.**\
> Thanks for the constructive comment. Based on the suggestion, we have added the comparative experiment that uses self-generated and self-filtered baseline data to perform supervised fine tuning (SFT). The results are shown in the following table, and have this table and corresponding discussion and analysis in **Tab. 2 and Sec. 4.2** **in the revised version**. From these results, we can see that: (1) The performance is slightly lower than that of the baseline itself, indicating that the quality of the generated images by the baseline is insufficient to meet the high-quality data requirements of the SFT training framework. An effective solution is to use high-quality manually labeled data; however, this would consume a significant amount of manpower and resources. (2) Our OSPA significantly outperforms the baseline because it does not rely on the quality of individual samples generated by the baseline model, but rather on the preference level between generated images. Moreover, our approach eliminates the need for manual annotation, as it autonomously computes rewards through the model's own components to steer preference learning. We have conducted experiments on four distinct baselines, and our method achieved varying degrees of performance improvement across all baselines, demonstrating its robustness and generalization capability.
>
> | Methods | Sim↑ | CLIP-I↑ | FLIP-I↑ |
> | :---- | :---- | :---- | :---- |
> | baseline | 45.65 | 57.82 | 54.58 |
> | \+ SFT | 42.60 | 56.69 | 54.24 |
> | **\+ OSPA (Ours)** | **53.89** | **60.59** | **60.51** |
>
> **Q3. Please clarify the exact strategy used for selecting the critical noise hyperparameter $\alpha$ for each of the four baselines. Was a single value universally effective, or was individual tuning necessary? If tuning was needed, how sensitive are the results to small variations in $\alpha$ for the different architectures?** \
> Thank you for the professional advice. We set the range of $\alpha$ values ​​for different baselines based on the following two criteria: (1) For the baseline with an image encoder (such as IP-Adapter, InstantID, and InfiniteYou) that is highly sensitive to faces, the $\alpha$ value should be set to a small range, such as (0.01\~0.05). This is because even slight noise perturbations can alter the identity embedding. (2) For the baseline with an image encoder (such as IP-AdapterPlus) that is not sensitive to faces, the $\alpha$ value should be set to a large range, such as (0.75\~3.00), thus enabling the generation of preference pairs to steer the model's self-preference optimization. For more discussion and analysis on $\alpha$ settings for image encoders with different face sensitivity levels, please refer to **Fig. 9 and Appendix C in the revised version**. In addition, we also provide the sensitivity analysis of the hyperparameter $\alpha$ for four baselines (IP-Adapter, InstantID, InfiniteYou, and IP-AdapterPlus) in **Fig. 8, 10, Sec.4.4, and  Appendix C in the revised version**. From these results, we can see that the hyperparameter $\alpha$ exhibits stable and good performance across different value ranges corresponding to different image encoders.

---

> > ### Comment · Reviewer_kjRG · 2025-11-28
> >
> > Thanks for the author's thoughtful response. I would like to maintain my original evaluation of this work.

---

> > > ### Author Response · Authors · 2025-11-28
> > >
> > > Thank you for the positive feedback and for maintaining the score. We appreciate your professional review and are available for any further questions.

---

### Official Review · Reviewer_y52x · 2025-10-30

**Soundness:** 2
**Presentation:** 3
**Contribution:** 2
**Rating:** 4
**Confidence:** 5

**Summary:**

This paper focuses on addressing limitations in identity-preserving text-to-image generation (e.g., supervised fine-tuning lacking feedback, RLHF relying on costly external resources) and proposes OSPA (Online Self-Preference Alignment), a plug-and-play framework. OSPA uses three core modules: self-preference sample generation, self-reward optimization, and online curriculum learning. Experiments have shown the effectiveness of the method. .

**Strengths:**

1. OSPA eliminates the need for expensive external reward models and high-quality curated preference datasets, which are required by existing methods like RLHF-based approaches. Instead, it leverages self-generated preference signals and intrinsic self-reward mechanisms, reducing costs and practical constraints .
2. As a flexible framework, OSPA can be seamlessly applied to multiple SOTA identity-preserving text-to-image models without modifying their core architectures. This broad compatibility makes it highly applicable to existing systems.
3. Extensive experiments show OSPA enhances identity preservation while maintaining or even improving visual quality, addressing the trade-off faced by many existing methods .
4. This paper is well-written and easy to follow.

**Weaknesses:**

1. OSPA operates on top of pre-trained identity-preserving text-to-image models, and its effectiveness is directly influenced by the quality of these underlying baselines. As explicitly stated in the paper, stronger baselines lead to larger performance gains, while weaker baselines limit the extent of improvement. This means OSPA cannot independently address the inherent flaws of poor-quality baseline models, which restricts its applicability.
2. The self-preference sample generation module relies on Gaussian noise perturbation to create preferred/unpreferred sample pairs. However, experiments show that increasing noise intensity significantly reduces facial identity similarity. This indicates that OSPA’s noise perturbation strategy requires careful parameter tuning. The lack of an adaptive noise adjustment mechanism makes it less robust.
3. For evaluation, this work relies on just 30 reference images from FFHQ and 40 prompts. The small scale and narrow scope of the datasets may limit the generalization of OSPA’s performance to real-world scenarios with more varied identity types and text prompts.
4. The experiments only validate OSPA under standard conditions. There is no testing on extreme scenarios, such as low-quality reference images (blurred, occluded, or low-light), complex text prompts, or cross-domain identity preservation (e.g., generating real-world photos from stylized images).

**Questions:**

1. OSPA operates on top of pre-trained identity-preserving text-to-image models, and its effectiveness is directly influenced by the quality of these underlying baselines. As explicitly stated in the paper, stronger baselines lead to larger performance gains, while weaker baselines limit the extent of improvement. This means OSPA cannot independently address the inherent flaws of poor-quality baseline models, which restricts its applicability.
2. The self-preference sample generation module relies on Gaussian noise perturbation to create preferred/unpreferred sample pairs. However, experiments show that increasing noise intensity significantly reduces facial identity similarity. This indicates that OSPA’s noise perturbation strategy requires careful parameter tuning. The lack of an adaptive noise adjustment mechanism makes it less robust.
3. For evaluation, this work relies on just 30 reference images from FFHQ and 40 prompts. The small scale and narrow scope of the datasets may limit the generalization of OSPA’s performance to real-world scenarios with more varied identity types and text prompts.
4. The experiments only validate OSPA under standard conditions. There is no testing on extreme scenarios, such as low-quality reference images (blurred, occluded, or low-light), complex text prompts, or cross-domain identity preservation (e.g., generating real-world photos from stylized images).

---

> ### Author Response · Authors · 2025-11-27
> **Official Comment by Authors (Part 1)**
>
> Thanks for your positive recognition (e.g.,**flexible framework**, **broad compatibility**, **well-written manuscript**). We appreciate the opportunity to address your valuable feedback as follows.
>
> **Response to Weaknesses & Questions**
> ---
>
> **W1 & Q1. OSPA operates on top of pre-trained identity-preserving text-to-image models, and its effectiveness is directly influenced by the quality of these underlying baselines. As explicitly stated in the paper, stronger baselines lead to larger performance gains, while weaker baselines limit the extent of improvement. This means OSPA cannot independently address the inherent flaws of poor-quality baseline models, which restricts its applicability.**\
> Thank you for your comments. Our OSPA is built upon pre-trained identity-preserving text-to-image models, as proposed as a plug-and-play framework that achieves identity-preserving generation without relying on external reward models or high-quality datasets. Like other self-improvement approaches, OSPA is inherently constrained by the capabilities of the base model. If the model lacks a certain ability initially, its own generations may not provide meaningful learning signals for improvement. In addition, our OSPA method does not aim to solve the inherent defects of the initial model, but rather guides the model to learn towards its own preferences through the generation of self-preference data and a self-reward mechanism. A promising approach to mitigating the inherent limitations of a model is to improve its initial performance through supervised fine-tuning, followed by further refining the model's self-preference using our OSPA method to achieve even better results. We have added this discussion and limitations **in Sec. 5 and Appendix C in the revised version**.
>
> **W2 & Q2. The self-preference sample generation module relies on Gaussian noise perturbation to create preferred/unpreferred sample pairs. However, experiments show that increasing noise intensity significantly reduces facial identity similarity. This indicates that OSPA’s noise perturbation strategy requires careful parameter tuning. The lack of an adaptive noise adjustment mechanism makes it less robust.**\
> Thank you for your valuable feedback. Our imprecise description of Fig. 6 may have led to potential misunderstandings. Note that the vertical axis in Fig. 6 represents similarity scores of noise-perturbed images rather than fully trained model performance. In fact, Fig. 6 demonstrates that adjusting the noise intensity significantly impacts face similarity while having minimal effects on background similarity and image-text alignment. It is precisely this sensitivity to face similarity that enables us to generate diverse facial preference data pairs by modulating noise intensity, which are subsequently used to train the policy model.
>
> Regarding the adaptive adjustment mechanism for $\alpha$, we propose two scenarios: First, for general image encoders that exhibit lower sensitivity to faces, we set a larger $\alpha$ value to sufficiently perturb ID embeddings. Second, for face-specific image encoders that demonstrate higher facial sensitivity, we assign a smaller $\alpha$ value to adjust ID embeddings. This dual adaptive mechanism eliminates the need for per-encoder $\alpha$ tuning \- we simply analyze the encoder's facial sensitivity level and randomly select a value according to the above cases to achieve self-preference data generation and self-reward training.
>
> Consequently, our method demonstrates remarkable generalization capability and can be seamlessly transferred to various ID-preserving text-to-image approaches. We validated $\alpha$’s robustness range for both encoder types using IP-Adapter and IP-Adapter Plus, as shown in **Fig. 8 in the revised version**. Experimental results confirm our method's significant robustness across different image encoders within certain $\alpha$ intervals. Additional analyses on $\alpha$ adaptation and robustness are provided in the revised **Sec. 4.4 and Appendix C**.

---

> ### Author Response · Authors · 2025-11-27
> **Official Comment by Authors (Part 2)**
>
> **W3 & Q3. For evaluation, this work relies on just 30 reference images from FFHQ and 40 prompts. The small scale and narrow scope of the datasets may limit the generalization of OSPA’s performance to real-world scenarios with more varied identity types and text prompts.** \
> Regarding the evaluation dataset, its size is consistent with the scale of datasets used in existing ID-preserving generation works (such as ID-Aligner, InfiniteYou). The following table shows the comparison results. It is noted that there are no evaluation datasets for IP-adapter and Instant ID regarding faces. Moreover, our reference images are carefully selected to cover diverse races, genders, and ages, representing a rich spectrum of identities. Our prompts encompass tasks from various scenarios, including stylization, actions, background, and clothing. We have included these detailed datasets **in Appendix B and Tab. 7** of the updated paper.
>
> | Methods | \# Reference | \# Prompt | \# Sample | \# Total Image |
> | :---- | :---- | :---- | :---- | :---- |
> | ID-Aligner | 20 | 40 | **5** | 4000 |
> | InfiniteYou | 15 | **200** | \- | 1497 |
> | **OSPA (Ours)** | **30** | 40 | 4 | **4800** |
>
> ##
>
> **W4 & Q4. The experiments only validate OSPA under standard conditions. There is no testing on extreme scenarios, such as low-quality reference images (blurred, occluded, or low-light), complex text prompts, or cross-domain identity preservation (e.g., generating real-world photos from stylized images).**\
> Thank you for raising this point. In response, we have conducted experiments covering precisely these scenarios. Specifically, we show the visualization results of IP-Adapter \+ OSPA **in Fig. 13, 14, and 15 in the revised version**. The results demonstrate that our OSPA maintains robust performance even under certain extreme conditions. Due to the limitations of the pretrained model of baseline, we cannot avoid the problem of identity preservation in certain extreme cases, such as masked, low-light, and stylized images, even training with the OPSA framework. Like other self-improvement approaches, OSPA is inherently constrained by the capabilities of the base model, and we have added the limitations and discussion of our approach **in Appendix C in the revised version**.

---

### Official Review · Reviewer_rPWF · 2025-11-01

**Soundness:** 3
**Presentation:** 2
**Contribution:** 3
**Rating:** 4
**Confidence:** 4

**Summary:**

The paper introduces a novel method, Online Self-Preference Alignment (OSPA), designed to overcome the requirement for human input or external reward models in current methods for identity-preserving text-to-image generation. By utilizing only the policy model for alignment, OSPA demonstrates commendable performance.

**Strengths:**

1. The three core components of the proposed OSPA method are well-motivated and specifically designed to address distinct problems, although some of them appear incremental.

2. The visualization and experimental results presented are compelling.

**Weaknesses:**

#### Disadvantage/Limitations

##### Minor Limitations

###### a) Typographical and Formula Errors

1. In Algorithm 1, line 8 (or line 227 of the manuscript), the term $\mathrm{sim}(\bm{v}_{\mathrm{ref}}, \mathcal{E}_{I}(\bm{x}_{\mathrm{gen}}^{u_{i}}))$ appears to be a typo. I suspect it should instead be $\mathrm{sim}(\bm{v}_{\mathrm{ref}}, \mathcal{E}_{I}(\bm{x}_{\mathrm{gen}}^{p_{i}}))$ to align with the intended logic of comparing the generated preferred image with the reference vector.

2. If Equation (6) is correct as written, the term involving $\epsilon_{\mathrm{ref}}$ cannot function as a regularizer for the parameter $\epsilon_{\theta}$ as they appear disconnected. I suggest the authors verify if the second term should be $\|\epsilon_{\theta}(\mathbf{x}_{t}^{i}, t) - \epsilon_{\mathrm{ref}}(\mathbf{x}_{t}^{i}, t)\|^{2}_{2}$. Furthermore, the bracket placement seems incorrect; the scaling factor $A_{i}$ should likely only multiply the first term of the loss function.

###### b) Writting

1. The paper references $\mathrm{DDIM}_{\mathrm{sample}}$ in Equations (3) and (4). While DDIM is a common method, the authors must provide the detailed algorithm and corresponding settings in the Appendix for reproducibility (e.g., the time schedule, the exact number of sampling steps, and whether stochasticity is used). Similarly, Equation (5) references the similarity function, $\mathrm{sim}$. The authors must state how this similarity is calculated (e.g., cosine similarity or Euclidean distance).

##### Major Limitations

1. The precise configuration of the "noise identity" (specifically the scale factor $A_i$) is unclear, leading to questions about the critical initial **self-preference sample generation**. This initial step is vital, as subsequent Reinforcement Learning (RL) and online fine-tuning rely entirely on this synthetic dataset. I think the authors should provide an **ablation study** to explicitly validate the effect and necessity of the scale factor $\alpha$.

2. The experiments are exclusively conducted on facial datasets. Given that related work (e.g., IP-Adapter [1]) has validated identity-preserving techniques on diverse non-face datasets, the proposed OSPA method should also be tested on broader image categories to demonstrate its generalizability. Furthermore, I think OPSA should be compared against methods that use external reward models (like ID-Aligner [2]) or human-annotated data. Even a slight performance degradation is acceptable, as a direct comparison is necessary to validate the claim that the cost reduction (no external models/human data) justifies the trade-off.

[1] Ye, H., Zhang, J., Liu, S., Han, X., & Yang, W. (2023). IP-Adapter: Text compatible image prompt adapter for text-to-image diffusion models. arXiv preprint arXiv:2308.06721.
[2] Chen, W., Zhang, J., Wu, J., Wu, H., Xiao, X., & Lin, L. (2024). ID-Aligner: Enhancing identity-preserving text-to-image generation with reward feedback learning. arXiv preprint arXiv:2404.15449.

**Questions:**

1. Regarding the use of DDIM, since the sampler is typically deterministic, I am confused by Figure 3(a), which shows a diverse set of images in the preference group generated from a single image and text input. Can the authors clarify where the randomness is introduced during the sampling process (apart from the initial noise injection for the preference/unpreference images)? Additionally, while the paper specifies the timesteps used for gradient updates, the total sampling steps for DDIM is missing (e.g. 50 steps or 100 steps). Also, can the authors offer the memory cost for a single gradient descent step and the overall time cost for training the model.

2. The authors validate Gaussian noise over "salty noise," but there are many other viable perturbation strategies for the embedding space. If possible, could the authors explore and conduct experiments using alternative perturbation methods, such as randomly replacing parts of the embedding vector with Gaussian noise or randomly swapping values within the embedding?

---

> ### Author Response · Authors · 2025-11-27
> **Official Comment by Authors (Part 1)**
>
> Thanks for your in-depth review, endorsement of our work (e.g., **good motivation** and **compelling results**) and constructive advice. Below, we provide detailed responses to each of your points.
>
>
>
> **Response to Weaknesses**
> ---
>
> **W1. In Algorithm 1, line 8 (or line 227 of the manuscript), the term $\mathrm{sim}(\mathbf{v}\_{\mathrm{ref}}, \mathcal{E}\_{I}(\mathbf{x}\_{\mathrm{gen}}^{u\_{i}}))$ appears to be a typo. I suspect it should instead be $\mathrm{sim}(\mathbf{v}\_{\mathrm{ref}}, \mathcal{E}\_{I}(\mathbf{x}\_{\mathrm{gen}}^{p\_{i}}))$ to align with the intended logic of comparing the generated preferred image with the reference vector.**\
> Thank you for carefully reading and pointing out this formula error. We have revised the formula error on line 8 in Algorithm 1, and double-checked all symbols and formulas highlighted in blue in the revised version.
>
> **W2. If Equation (6) is correct as written, the term involving cannot function as a regularizer for the parameter as they appear disconnected. I suggest the authors verify if the second term should be $|\epsilon\_{\theta}(\mathbf{x}\_{t}^{i}, t) - \epsilon\_{\mathrm{ref}}(\mathbf{x}\_{t}^{i}, t)|^{2} _ {2}$. Furthermore, the bracket placement seems incorrect, the scailing factor $A_{i}$ should likely only multiply the first term of the loss function.**\
> Sorry for the incorrect placement of parentheses in Eq. (6). We have corrected it with the proper bracket placement highlighted in blue in the revised version. As a result, the normalized reward $A_{i}$ is multiplied by both the first and second terms, making the regularization term take the same form as in GPO, thus allowing $A_{i}$ to correctly steer the optimization direction.
>
> **W3. The paper references in Equations (3) and (4). While DDIM is a common method, the authors must provide the detailed algorithm and corresponding settings in the Appendix for reproducibility (e.g., the time schedule, the exact number of sampling steps, and whether stochasticity is used). Similarly, Equation (5) references the similarity function, sim. The authors must state how this similarity is calculated (e.g., cosine similarity or Euclidean distance).**\
> We sincerely thank you for pointing out the missing implementation details of Eq. (3), (4) and (5). We have added specific implementation details of Eq. (3) and (4) for each baseline model based on our OSPA framework, and the formula of Eq. (5) for calculating cosine similarity in Eq. (9) in **Appendix B in the revised version (highlighted in blue)**.
>
> **W4. The precise configuration of the "noise identity" (specifically the scale factor $A_{i}$) is unclear, leading to questions about the critical initial self-preference sample generation. This initial step is vital, as subsequent Reinforcement Learning (RL) and online fine-tuning rely entirely on this synthetic dataset. I think the authors should provide an ablation study to explicitly validate the effect and necessity of the scale factor $\alpha$.**\
> For the issues regarding unclear configuration of noise identity $\alpha$, we have provided detailed analysis highlighted in blue in **Fig. 8, 9, 10,** **Sec. 4.4, Appendix C in the revised version**. Based on the sensitivity of the baseline image encoder to faces, there are two setting mechanisms: smaller $\alpha$ values are used for the baselines, which the image encoder is sensitive to faces due to a small coefficient $\alpha$ can alter the identity embedding, while the larger $\alpha$ values are used for the baselines that the image encoder is less sensitive to faces. Regarding the ablation study of the scale factor $\alpha$, we have provided and discussed **in Tab. 3** and **Sec. 4.3 in the revised version**.

---

> ### Author Response · Authors · 2025-11-27
> **Official Comment by Authors (Part 2)**
>
> **W5-(1). The experiments are exclusively conducted on facial datasets. Given that related work (e.g., IP-Adapter \[1\]) has validated identity-preserving techniques on diverse non-face datasets, the proposed OSPA method should also be tested on broader image categories to demonstrate its generalizability.**
>
> **\[1\] Ye, H., Zhang, J., Liu, S., Han, X., & Yang, W. (2023). IP-Adapter: Text compatible image prompt adapter for text-to-image diffusion models. arXiv preprint arXiv:2308.06721.**\
> Thanks for your advice. We have added broader identity-preserving visualization results (**Fig. 16**) along with corresponding discussions in **Appendix C**. The results show that our method has certain limitations in non-face identity preservation. This is because our pre-trained model is face-based, and we use a large amount of face data for augmentation training. Therefore, we achieve significant performance in face identity fidelity (see the **Tab.1**). It is worth mentioning that our OSPA is a plug-and-play framework, and we will transfer it to other non-face identity fidelity tasks in future work. For more discussion and analysis, please refer to **Appendix C**.
>
> **W5-(2). Furthermore, I think OPSA should be compared against methods that use external reward models (like ID-Aligner \[2\]) or human-annotated data. Even a slight performance degradation is acceptable, as a direct comparison is necessary to validate the claim that the cost reduction (no external models/human data) justifies the trade-off.**
>
> **\[2\] Chen, W., Zhang, J., Wu, J., Wu, H., Xiao, X., & Lin, L. (2024). ID-Aligner: Enhancing identity-preserving text-to-image generation with reward feedback learning. arXiv preprint arXiv:2404.15449.**\
> Thanks for the constructive comment. Based on the suggestion, we have added the comparison results (as shown in the following table) and discussion **in Tab. 2** and **Sec. 4.2 in the revised version**. The results show that our OSPA achieves better identity fidelity than ReFL (ID-Aligner) across most metrics, indicating that our OSPA can achieve identity-preserving generation without relying on external reward models or high-quality datasets.
>
> | Methods | Sim↑ | CLIP-I↑ | FLIP-I↑ |
> | :---- | :---- | :---- | :---- |
> | baseline | 45.65 | 57.82 | 54.58 |
> | \+ ReFL (ID-Aligner) | 51.32 | **60.90** | 59.93 |
> | **\+ OSPA (Ours)** | **53.89** | 60.59 | **60.51** |
>
> **Response to Questions**
> ---
>
> **Q1-(1). Regarding the use of DDIM, since the sampler is typically deterministic, I am confused by Figure 3(a), which shows a diverse set of images in the preference group generated from a single image and text input. Can the authors clarify where the randomness is introduced during the sampling process (apart from the initial noise injection for the preference/unpreference images)?**
> Thanks for your meaningful questions. While DDIM is indeed deterministic given a fixed initial latent noise, the diversity in Fig. 3(a) arises precisely from sampling different initial noises for each generated image, even though the input image and text prompt are the same. In addition, we assign distinct seeds in the online curriculum process to the generated images.
>
>
> **Q1-(2). Additionally, while the paper specifies the timesteps used for gradient updates, the total sampling steps for DDIM is missing (e.g. 50 steps or 100 steps). Also, can the authors offer the memory cost for a single gradient descent step and the overall time cost for training the model.**\
> Regarding the total sampling steps for DDIM, we have introduced it in Appendix B. Specifically, we set **30 steps** for IP-Adapter, IP-AdapterPlus, and InstantID, and **10 steps** for InfiniteYou, which are enhanced by our OSPA. In addition, the memory cost and overall training time cost are reported in the following table.(Using a single 80G A100.)
>
> | Methods | Memory Cost (G) | GPU Training Times (H) |
> | :---- | :---- | :---- |
> | IP-Adapter \+ OSPA | 42 | 46 |
> | IP-Adapter-plus \+ OSPA | 45 | 47 |
> | InstantID \+ OSPA | 75 | 120 |
> | InfiniteYou \+ OSPA | 78 | 168 |
>
> **Q2. The authors validate Gaussian noise over "salty noise," but there are many other viable perturbation strategies for the embedding space. If possible, could the authors explore and conduct experiments using alternative perturbation methods, such as randomly replacing parts of the embedding vector with Gaussian noise or randomly swapping values within the embedding?**\
> Thank you for your comments and suggestions. We have added more comparisons of perturbation strategies, including randomly replacing parts of the embedding vector with Gaussian noise and randomly swapping values within the embedding. In **Appendix C** and **Fig. 11 in the revised version**, we have discussed that the perturbation strategy of embedding Gaussian noise in all embeddings is more suitable for guiding the generation of data pairs with preferences compared to the two perturbation methods mentioned above.

---

### Official Review · Reviewer_yLx2 · 2025-11-02

**Soundness:** 2
**Presentation:** 2
**Contribution:** 2
**Rating:** 4
**Confidence:** 3

**Summary:**

This paper proposes Online Self-Preference Alignment (OSPA), a plug-and-play framework for enhancing identity-preserving text-to-image generation. Instead of relying on external reward models or high-quality curated datasets (limitations of existing supervised fine-tuning (SFT) and reinforcement learning with human feedback (RLHF) approaches), OSPA achieves identity preservation via self-generated preference signals.

**Strengths:**

1. The paper repurposes generative models’ stochasticity to generate self-preference pairs via Gaussian noise on embeddings, eliminating the need for external annotations or models.
2. As a plug-and-play framework, the method adapts to 4 SOTA baselines (e.g., IP-Adapter, InstantID) by only updating adapters/projectors, ensuring easy deployment.
3. Multiple metrics (face similarity, CLIP-I, FLIP-I) and ablation studies (noise, online/offline updates) confirm the reliability of OSPA’s performance.

**Weaknesses:**

1. The paper compares OSPA to SFT-based baselines but not to RLHF/DPO methods (e.g., ID-Aligner, Diffusion-DPO) on the same dataset. Without this, readers cannot fully assess whether OSPA’s gains are due to its design or simply the choice of baselines.
2. While Fig. 6 shows noise intensity impacts face similarity, the paper does not explain how to choose α (noise coefficient) for different models. For example, IP-Adapter uses α=0.025, InstantID uses α=0.04—what guides this choice?
3. The GPO loss (Eq. 6) references "shifted timestep sampling (H) from SD3" but provides no details on how H is configured (e.g., timestep range, sampling frequency). Without this, researchers cannot replicate the loss function accurately.

**Questions:**

1. What are OSPA’s main failure cases? For example, does it struggle with: (a) reference images with occlusions (e.g., glasses, masks)? (b) prompts with extreme style changes (e.g., "person as a cartoon")?
2. In online curriculum learning, continuously updating the policy with the optimized target model may cause distribution drift. Have the authors theoretically proven the convergence conditions of the OSPA framework? In experiments, how is it determined that the model has reached a stable state? Is there a risk of decreased identity fidelity in the later training stages due to over-exploration?

---

> ### Author Response · Authors · 2025-11-27
> **Official Comment by Authors (Part 1)**
>
> Thank you for your encouraging appreciation (e.g., **no external annotations or models required**, **easy deployment**, and **reliable performance**). Below, we provide detailed responses and clarifications corresponding to each of your points.
>
> **Response to Weaknesses**
> ---
>
> **W1. The paper compares OSPA to SFT-based baselines but not to RLHF/DPO methods (e.g., ID-Aligner, Diffusion-DPO) on the same dataset. Without this, readers cannot fully assess whether OSPA’s gains are due to its design or simply the choice of baselines.**
> Thank you for your insightful comments and suggestions. We focused on comparing our OSPA with SFT-based baselines because: (1) Most identity-preserving image generation methods are based on supervised fine-tuning techniques; (2) The code of ID-Aligner, which is the only one RLHF method for identity-preserving image generation, is not publicly available, making a direct comparison impossible; (3) The DPO methods such as Diffusion-DPO remain underexplored for ID-preserving generation. However, we fully agree that such a comparison is valuable and will strengthen the evaluation of our OSPA’s effectiveness. Therefore, we train the same baseline model using ReFL (the training framework in  ID-Aligner) and DPO (the training framework in Diffusion-DPO), and compare them with our OSPA framework, as shown in the table below. It is clear that our framework significantly outperforms ReFL and DPO. For more results and discussion, please refer to **Sec. 4.2 (highlighted in blue) in the revised version.**
>
> | Methods | Sim↑ | CLIP-I↑ | FLIP-I↑ |
> | :---- | :---- | :---- | :---- |
> | baseline | 45.65 | 57.82 | 54.58 |
> | \+ ReFL | 51.32 | **60.90** | 59.93 |
> | \+ DPO | 50.29 | 59.36 | 57.52 |
> | **\+ OSPA (Ours)** | **53.89** | 60.59 | **60.51** |
>
> **W2. While Fig. 6 shows noise intensity impacts face similarity, the paper does not explain how to choose $\alpha$ (noise coefficient) for different models. For example, IP-Adapter uses $\alpha$=0.025, InstantID uses $\alpha$=0.04—what guides this choice?**\
> Thank you for your insightful and thought-provoking question. The setting of the noise intensity coefficient $\alpha$ is indeed model-dependent. However, we tailor the range of the noise intensity coefficient $\alpha$ to the image encoder's inherent sensitivity (see Figure 3 (a)) to faces. Specifically, (1) If the image encoders (such as IP-Adapter, InstantID, and InfiniteYou) have good encoding capabilities for face images, a slight perturbation (small coefficient $\alpha$) can alter the identity embedding, resulting in generated faces that deviate from the original identity. Conversely, if the $\alpha$ is too large, it may corrupt the face embedding, preventing the generation of a complete face image and resulting in the data groups (see in the Figure 3 (a)) lacking effective preferences; (2) If the image encoder (such as IP-AdapterPlus) has weak encoding capabilities for face images, a strong perturbation (large coefficient $\alpha$) is needed to perturb the identity embedding for generating face images with identity differences from the original face images to construct effective preference data pairs. Conversely, if the $\alpha$ coefficient is too small, it cannot effectively modify the identity information in the image embedding, thus failing to generate preference data pairs with some differences, affecting subsequent preference training. The corresponding analysis is provided in **Fig. 9** and **Appendix C**. In addition, to verify the robustness of $\alpha$ to the two type image encoder using different baselines, we conducted the sensitivity analysis for four baselines (IP-Adapter, IP-AdapterPlus, InstantID, InfiniteYou) enhanced with our OSPA. The corresponding results and analysis are provided in **Fig. 8, and Fig. 10, Sec. 4.4, and Appendix C.**
>
> **W3. The GPO loss (Eq. 6\) references "shifted timestep sampling (H) from SD3" but provides no details on how H is configured (e.g., timestep range, sampling frequency). Without this, researchers cannot replicate the loss function accurately.**\
> Thanks for your advice. We follow the GPO\[1\] and set $\beta=1$ for the shifted timestep sampling strategy $H$. We added more details, highlighted in blue in **Appendix B in the revised version**.
> \[1\] Chen R, Lin W, Zhang Y, et al. Towards Self-Improvement of Diffusion Models via Group Preference Optimization\[J\]. arXiv preprint arXiv:2505.11070, 2025\.

---

> ### Author Response · Authors · 2025-11-27
> **Official Comment by Authors (Part 2)**
>
> **Response to Questions**
> ---
>
> **Q1. What are OSPA’s main failure cases? For example, does it struggle with: (a) reference images with occlusions (e.g., glasses, masks)? (b) prompts with extreme style changes (e.g., "person as a cartoon")?**\
> Thanks for your interesting questions. We have added some extreme cases (**shown in Fig. 13, 14, 15, and 16**) along with corresponding discussions in **Appendix C**. Our current version of OSPA struggles to improve identity fidelity in masked, low-light, and stylized images. One possible reason is that the pre-trained model of baseline was trained on real images with minimal occlusion, no significant lighting issues. Our OSPA method cannot address the inherent flaws of the baseline, but we can improve the model's self-preference level.
>
> **Q2-(1). In online curriculum learning, continuously updating the policy with the optimized target model may cause distribution drift. Have the authors theoretically proven the convergence conditions of the OSPA framework?** \
> Thank you for pointing out this concern and questions. A full theoretical proof of convergence for an online, non-stationary, self-play reinforcement learning remains significantly challenging due to the interdependent dynamic optimization of the policy model and the online curriculum learning, especially in the field of image generation, such as our OSPA framework. However, this online manner has been used in reinforcement learning frameworks, such as \[1\], \[2\] and \[3\]. In addition, we will provide a complete theoretical proof in our future work.
> \[1\] Hamadanian P, Nasr-Esfahany A, Schwarzkopf M, et al. Online Reinforcement Learning in Non-Stationary Context-Driven Environments\[C\]//The Thirteenth International Conference on Learning Representations. 2025\.
> \[2\] Bai C, Zhang Y, Qiu S, et al. Online Preference Alignment for Language Models via Count-based Exploration\[C\]//The Thirteenth International Conference on Learning Representations. 2025\.
> \[3\] Gupta R, Sullivan R, Li Y, et al. Robust Multi-Objective Preference Alignment with Online DPO\[C\]//Proceedings of the AAAI Conference on Artificial Intelligence. 2025, 39(26): 27321-27329.
>
>
>
> **Q2-(2). In experiments, how is it determined that the model has reached a stable state? Is there a risk of decreased identity fidelity in the later training stages due to over-exploration?**\
> We fully acknowledge the reviewer's valid concern regarding stability, which is indeed an important aspect. Given limited time resources, we have successfully demonstrated the effectiveness and superiority of the proposed method. The current training cycles are sufficient to reveal its potential and key characteristics. We consider convergence stability an achievable engineering optimization problem under more abundant resources. Due to training time constraints, our model has not yet reached a fully stable state at the steps we set. Accordingly, we have explicitly stated this limitation **in Appendix C in revised version** and provide the loss curves during training.
>
> Since our OSPA framework is trained on its own generated data, it inherently suffers from reduced diversity and risks eventual model collapse. If the model reaches training stability and training continues, its own generations may not provide meaningful learning signals for improvement. Due to training time constraints, our current experimental setup has not yet encountered a scenario with decreased identity fidelity from over-exploration. We acknowledge this as a valuable point and will prioritize investigating this risk in future work with extended training, more details are provided **in Appendix C in revised version**.

---

### Author Response · Authors · 2025-11-30
**Official Comment by Authors (Summary)**

**Dear Area Chairs, Senior Area Chairs, and Program Chairs,**

We thank the reviewers for their feedback, recognizing our work as **reliable performance** (yLx2), **good motivation** (rPWF), **flexible framework** (y52x), and **well-structured manuscript** (kjRG). Below, we address their concerns.

**yLx2(W1), rPWF(W5-(2)), and kjRG(W2 & Q2): Comparing OSPA with SFT, RLHF, and DPO frameworks**.\
The following table demonstrates OSPA's strong performance. For more analysis, please refer to **Sec.4.2**.
|Methods|Sim↑|CLIP-I↑|FLIP-I↑|
|-|-|-|-|
|baseline|45.65|57.82|54.58|
|\+SFT|42.60|56.69|54.24|
|\+ReFL|51.32|**60.90**|59.93|
|\+DPO|50.29|59.36|57.52|
|**\+OSPA(ours)**|**53.89**|60.59|**60.51**|

**yLx2(W2), rPWF(W4), y52x(W2 & Q2), and kjRG(W3 & Q3): Lack of setting, sensitivity analysis, and the ablation study of $\alpha$, and misleading Fig.6**.\
The $\alpha$ is set based on the face sensitivity of the baseline’s image encoder: small values (0.01\~0.05) for face-sensitive encoders (e.g., IP-Adapter) and large values (0.75\~3.00) for less sensitive ones (e.g., IP-AdapterPlus). We have provided detailed analysis in **Fig.9 and Appendix C**, sensitivity analysis in **Fig.8, 10, Sec.4.4, and Appendix C**, and an ablation study in **Tab.3 and Sec.4.3**, demonstrating $\alpha$'s robustness. Fig.6, with corrected caption compares the sensitive perturbation regions, precisely as our intended.

**yLx2(W3): Unclear details of GPO loss**.\
We have added more details in **Appendix B**.

**yLx2(Q1), rPWF(W5-(1)), and y52x(Q4 & W4): Validation of OSPA on failure cases and extreme scenarios**.\
We have validated OSPA on some failure cases and extreme scenarios, with results and discussion provided in **Fig.13~16 and Appendix C**.

**yLx2(Q2): Theoretical proof of convergence, determination of steady state, and over-exploration**.\
The theoretical convergence proof for OSPA, which is an online, non-stationary, self-play RL framework with interdependent policy model and online curriculum optimization, is highly challenging. We have presented empirical evidence in **Fig.12**, and noted that self-generated data may lead to reduced diversity and potential model collapse.

**rPWF(W1, W2, W3): Typographical and formula errors, and unclear details of Eq.3, 4 and 5**.\
We have revised all symbols and formulas, added specific details for Eq.3 and 4, and clarified that Eq.5 is the cosine similarity, as stated in **Eq.9** in the revised version.

**rPWF(Q1): Regarding the randomness of the sampling process, total sampling timesteps, memory cost, and training time**.\
We sample different initial noises and distinct seeds for each generated image in the online curriculum. Total sampling steps are provided in **Appendix B**. Memory cost and training time are reported in the following table.
|Methods|Memory Cost (G)|GPU Training Times (H)|
|-|-|-|
|IP-Adapter\+OSPA|42|46|
|IP-AdapterPlus\+OSPA|45|47|
|InstantID\+OSPA|75|120|
|InfiniteYou\+OSPA|78|168|

**rPWF(Q2): More perturbation strategies**.\
We have added additional strategies, as shown in **Fig.11 and Appendix C**, demonstrating that our perturbation strategy is most effective for generating preference pairs.

**y52x(W1 & Q1): OSPA's effectiveness and applicability**.\
OSPA is a plug-and-play and self-improvement framework without external reward models or datasets. Like other self-improvement methods, OSPA's capability is limited by the baseline. It guides self-preference learning, rather than fixing the baseline’s inherent flaws. We suggest pre-boosting the baseline via SFT before applying OSPA to mitigate inherent limitations. Additional discussion is provided in **Sec.5** and **Appendix C**.

**y52x(W3 & Q3): The small scale of evaluation datasets**.\
Our evaluation dataset is larger than those used in recent works (ID-Aligner, ACM MM 2024; InfiniteYou, ICCV 2025). Our reference images span diverse identities, and prompts cover diverse scenarios, demonstrating the OSPA's effectiveness across varied conditions.
|Methods|\# Reference|\# Prompt|\# Sample|\# Total Image|
|-|-|-|-|-|
|ID-Aligner|20|40|**5**|4000|
|InfiniteYou|15|**200**|\-|1497|
|**OSPA (Ours)**|**30**|40|4|**4800**|

**kjRG(W1 & Q1): Misleading framework of "Self-Preference" and "Self-Reward", and the ablation study on external encoder**.\
OSPA operates solely within each baseline’s components, including an image encoder, generating preference data via self-perturbation and using the image encoder as the reward model, without external data or models. Thus, OSPA’s performance is inherently bounded by the baseline. We validated OSPA across four baselines with different image encoders (**Tab.1**), demonstrating generalization. We further introduced **Tab.4** for the ablation study on an external encoder, confirming the effectiveness of our self-reward preference optimization mechanism.

In summary, we have provided detailed responses to all the reviewers' concerns and made thorough revisions in the revised version.

---

### Meta-Review · Area_Chair_jf1X · 2025-12-30

**Summary:**

The reviewers recognized the motivation of OSPA in eliminating the need for expensive external reward models but raised several concerns regarding the comparative evaluation and hyperparameter sensitivity. Specifically, reviewers pointed out the absence of comparisons against RLHF, DPO, and simple rejection sampling baselines to justify the method's advantage over established techniques. Other technical concerns involve the sensitivity and selection strategy for the noise hyperparameter $\alpha$ across different backbones to examine its robustness. Additionally, reviewers questioned the generalizability of the method due to the small evaluation dataset and lack of non-face or extreme scenario testing.

**Reviewer Concerns:**

The rebuttal addressed most critical empirical concerns by providing additional comparisons with ReFL, DPO, and SFT methods, demonstrating OSPA's superiority. The authors also clarified the selection mechanism for the noise parameter $\alpha$, and provided  sensitivity analyses to show robustness. Although the authors provided empirical evidence of stability, the theoretical proof for the convergence of the non-stationary online optimization remains unresolved. Additionally, the limitation that OSPA cannot correct inherent flaws in the baseline model (e.g., handling occlusions) remains an important concern regarding its generalizability.

**Reviewer Scores:**

Reviewer kjRG explicitly stated they would maintain their score of 6, appreciating the response but finding the original assessment regarding the reliance on the frozen encoder still valid. Reviewers yLx2, rPWF, and y52x (all scored 4) would likely not raise their scores much in spite of some added experiments; the theoretical gaps and the method's dependence on baseline quality limit the broader impact. Specifically, Reviewer y52x's concern about the method's inability to address inherent baseline flaws was acknowledged by the authors but confirmed as a limitation

---

### Decision · Program_Chairs · 2026-01-26

Reject